# Anxious individuals shift emotion control from lateral frontal pole to dorsolateral prefrontal cortex

Bob Bramson [1,2,3] ✉, Sjoerd Meijer [1,3], Annelies van Nuland[1], Ivan Toni[1,4] & Karin Roelofs [1,2,4]

Anxious individuals consistently fail in controlling emotional behavior, leading to excessive avoidance, a trait that prevents learning through exposure. Although the origin of this failure is unclear, one candidate system involves control of emotional actions, coordinated through lateral frontopolar cortex (FPl) via amygdala and sensorimotor connections. Using structural, functional, and neurochemical evidence, we show how FPl-based emotional action control fails in highly-anxious individuals. Their FPl is overexcitable, as indexed by GABA/glutamate ratio at rest, and receives stronger amygdalofugal projections than non-anxious male participants. Yet, high-anxious individuals fail to recruit FPl during emotional action control, relying instead on dorsolateral and medial prefrontal areas. This functional anatomical shift is proportional to FPl excitability and amygdalofugal projections strength. The findings characterize circuit-level vulnerabilities in anxious individuals, showing that even mild emotional challenges can saturate FPl neural range, leading to a neural bottleneck in the control of emotional action tendencies.

Anxiety disorders are highly prevalent and difficult to treat. This difficulty primarily stems from excessive avoidance of feared situations which prevents learning through exposure[1]. The ability to override these strong avoidance tendencies in favor of alternative actions involves a flexible action-selection process, known to rely on a distributed circuit revolving around the lateral frontopolar cortex (FPl) interacting with posterior parietal cortex, sensorimotor cortex (SMC), and amygdala[2–7]. When people are faced with the challenge of overriding automatic emotional behaviors, such as social avoidance tendencies, the FPl coordinates that distributed circuit to guide emotionally-adaptive behavior[5]. The FPl involvement is mechanistically causal as well as clinically consequential: emotional action selection fails after FPl interference[8], and variation in FPl recruitment predicts resilience against the development of emotional disorders later in life[9]. Here, we build on those insights to identify functional-, structural-, and neurochemical properties of FPl that explain variation

in the implementation of neural control over emotional behavior between high-anxiety individuals and their non-anxious peers.

Contemporary models of anxiety, based on rodent studies, have shown how hippocampal-amygdala afferents to agranular medial frontal areas drive avoidance of threatening situations and fear-like behaviors[10,11], whereas recurrent medial frontal signals in the same circuit reduce threat-responses and allow approach behavior[11–14]. However, extension of those rodent-based insights to human anxiety disorders has proven difficult[15,16], reflected in disappointing progress in the development of novel treatments for anxiety disorders[17]. Those translational efforts face a major challenge in the expansion of human granular prefrontal cortex, as compared to non-human models[18,19]. For instance, the neuronal organization and connectivity profile of the human FPl have no homologue in rodents nor other primates[20–23]. More precisely, the human FPl has access to both medial and lateral cortical circuits through its extensive connections with other frontal, parietal

[1]Donders Institute for Brain, Cognition and Behavior, Centre for Cognitive Neuroimaging, Radboud University Nijmegen, 6525 EN Nijmegen, The Netherlands. [2]Behavioral Science Institute (BSI), Radboud University Nijmegen, 6525 HR Nijmegen, The Netherlands. [3]These authors contributed equally: Bob Bramson, Sjoerd Meijer. [4]These authors jointly supervised this work: Ivan Toni, Karin Roelofs. ✉e-mail: bob.bramson@donders.ru.nl

and temporal association areas. In addition, human FPl has direct access to information coming from the amygdala via the amygdalo-fugal bundle[4,20]. In contrast, macaque prefrontal cortex does not share a region homologous to human FPl, and its amygdalae project mainly to medial but not lateral prefrontal regions[24,25]. Accordingly, recent work has suggested that FPl is involved in selecting emotional actions by influencing neural activity in sensorimotor cortices when different alternative options are available[5]. Critically, recruitment of FPl when controlling emotional behavior fails in patients with emotional disorders[26], and is a long term resilience factor against the development of post-traumatic stress symptoms[9]. Based on these findings, we reasoned that aberrant FPl recruitment might account for the difficulties experienced by individuals with anxiety in situations where they need to control emotional action tendencies. This study is set to understand whether and how FPl function explains altered control of emotional action tendencies in individuals with anxiety.

We combined Magnetic Resonance Spectroscopy (MRS), Diffusion Weighted Imaging (DWI), and functional MRI to capture neurochemical, structural, and functional properties of FPl during emotional action control. Examining those properties jointly is essential to explain variation in behavioral and neural indices of emotional-action control in anxious participants[2,8,27]. We exposed participants selected for high-anxiety to a mild emotional challenge requiring control over their emotional action tendencies, and compared that group to an existing age-matched dataset drawn from the general male population (in short, non-anxious peers[6]). The two groups achieved comparable success in emotional-action control, yet the anxious participants solved that challenge using Brodmann areas (BA) 9/46d (dorsolateral prefrontal cortex; dlPFC) and anterior cingulate cortex (ACC), rather than FPl as did the non-anxious peers. This anxiety-related shift in the frontal circuit supporting emotional-action control was complemented by neurochemical and structural differences in the FPl of the two groups. Namely, the FPl of anxious participants had higher neuronal excitability, as indexed by GABA/Glutamate ratio, and

stronger amygdalofugal projections, as indexed by MR-tractography. Furthermore, stronger amygdala connections, in the context of reduced FPl neuronal responsivity, significantly accounted for the anxiety-related shift towards those alternative control circuits in the frontal lobe. Together, these findings identify a circuit-level neural vulnerability in anxious individuals, opening the way to targeted interventions for enhancing control of emotional action tendencies[6].

## Results

Participants were selected for high anxiety (high-anxiety group; N = 52; 14 males, score > 30 on the Liebowitz Social Anxiety Scale, LSAS, Fig. 1A, B). They were compared to a convenience control group of participants recruited and reported on earlier[6] who were not selected based on anxiety scores (non-anxious peers; N = 41; all males). Comparison of these groups on an independent metric of trait anxiety (State Trait Anxiety Inventory, STAI, Y-2), showed that the high-anxiety group was indeed more anxious as compared to their non-anxious peers, t(84) = 5.5, p < 0.001, Fig. 1B.

### Anxious participants control emotional actions by relying on dorsolateral prefrontal cortex rather than FPl

Both groups performed a social approach-avoidance task (Fig. 2A) in which they approached or avoided happy and angry faces by pulling or pushing a joystick. Previous work has shown that approaching angry and avoiding happy faces requires control over automatic action tendencies to approach appetitive and avoid aversive situations[3,8]. Accordingly, participants made more correct responses when approaching happy and avoiding angry faces; the congruent condition (M = 96.3 %, std = 2.9) as compared to approaching angry and avoiding happy faces; the incongruent condition (M = 94.3%, std = 4.4), Fig. 2B. This resulted in a main effect of congruency on error rates across groups, as assessed by means of Bayesian mixed effects models: b = 0.198 CI [0.1 0.29]. In those analyses effects are considered significant if the Credible Interval (CI) does not include 0. There were no group differences in those behavioral congruency-effects in the mixed effect model; b = 0.03 [−0.06 0.12] with moderate evidence for the absence of an effect of group on behavioral congruency resulting from a follow-up Bayesian t-test; Bayes Factor (BF$_{01}$) = 4.3.

fMRI measurements obtained during task performance across groups (Fig. 2C) show that controlling emotional action tendencies increased activity in bilateral FPl [32 54 4; −30 56 8] and decreased activity in bilateral sensorimotor cortex [42 −26 68; −32 −26 68]. Full list of coordinates is described in supplementary materials (supplementary tables 1–3). These effects confirm previous independent fMRI findings[28,29] and isolate a spatially distributed neural circuit supporting the control of automatic emotional action tendencies and selection of affect-incongruent alternative actions. Separating this analysis for non-anxious and high-anxious participants showed significant FPl activation in non-anxious (already reported in ref. 6), but no statistically reliable neural congruency effect in the FPl for high-anxious participants, suggesting that they might rely less on FPl for control, Fig. 2C. To assess this possibility, we used Bayesian t-test to clarify potential absence of FPl recruitment in high-anxious individuals in the specific FPl territory recruited in healthy controls (Fig. 2C black circles). The Bayesian t-test confirmed this observation, providing moderate evidence for the absence of this effect in the high-anxiety group, BF$_{01}$ = 4.2. There was no interaction between group (high vs non-anxious) and neural congruency in FPl when correcting for voxels across the whole frontal cortex.

A second finding of this study concerns between-groups differences evoked during emotional-action control. High-anxiety participants had stronger neural activity for incongruent versus congruent trials in BA area 8B / area 9/ area 46D [24 30 34] as compared to their non-anxious peers, Fig. 2D, corrected for multiple comparisons over all voxels in the frontal lobe. This suggests that anxious individuals shift

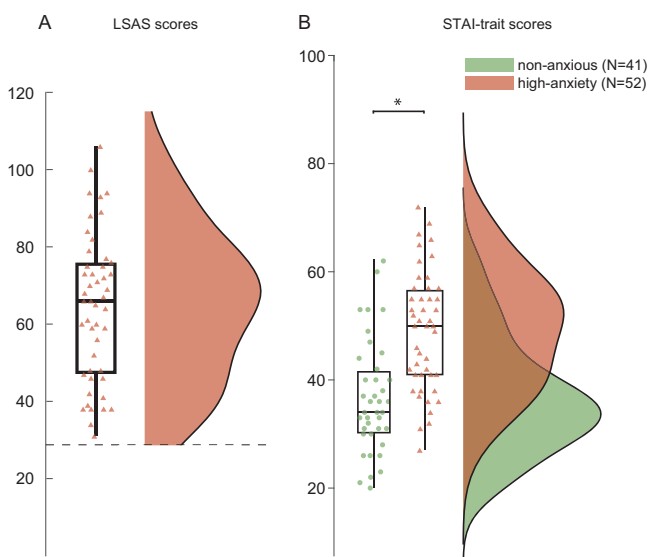

**Fig. 1 | Selection of high-anxious participants and group difference.**
**A** Participants in the high-anxious group (N = 52) were selected to have an Liebowitz Social Anxiety Scale (LSAS) score > 30 (dashed line), achieving a balance between sensitivity and specificity for detecting anxiety related disorders[97]. **B** LSAS-based selection resulted in a between groups difference in trait anxiety: t(84) = 5.5, p < 0.001, one-sided test, as independently indexed through the State Trait Anxiety Inventory (STAI; Y-2: trait anxiety). Non-anxious individuals: green circles; high-anxious individuals: red triangles. Asterix shows significant (p < 0.001) differences between conditions. Source data are provided as Source data file. Boxplot represent mean and 25% percentiles. Lines extend toward maximum values.

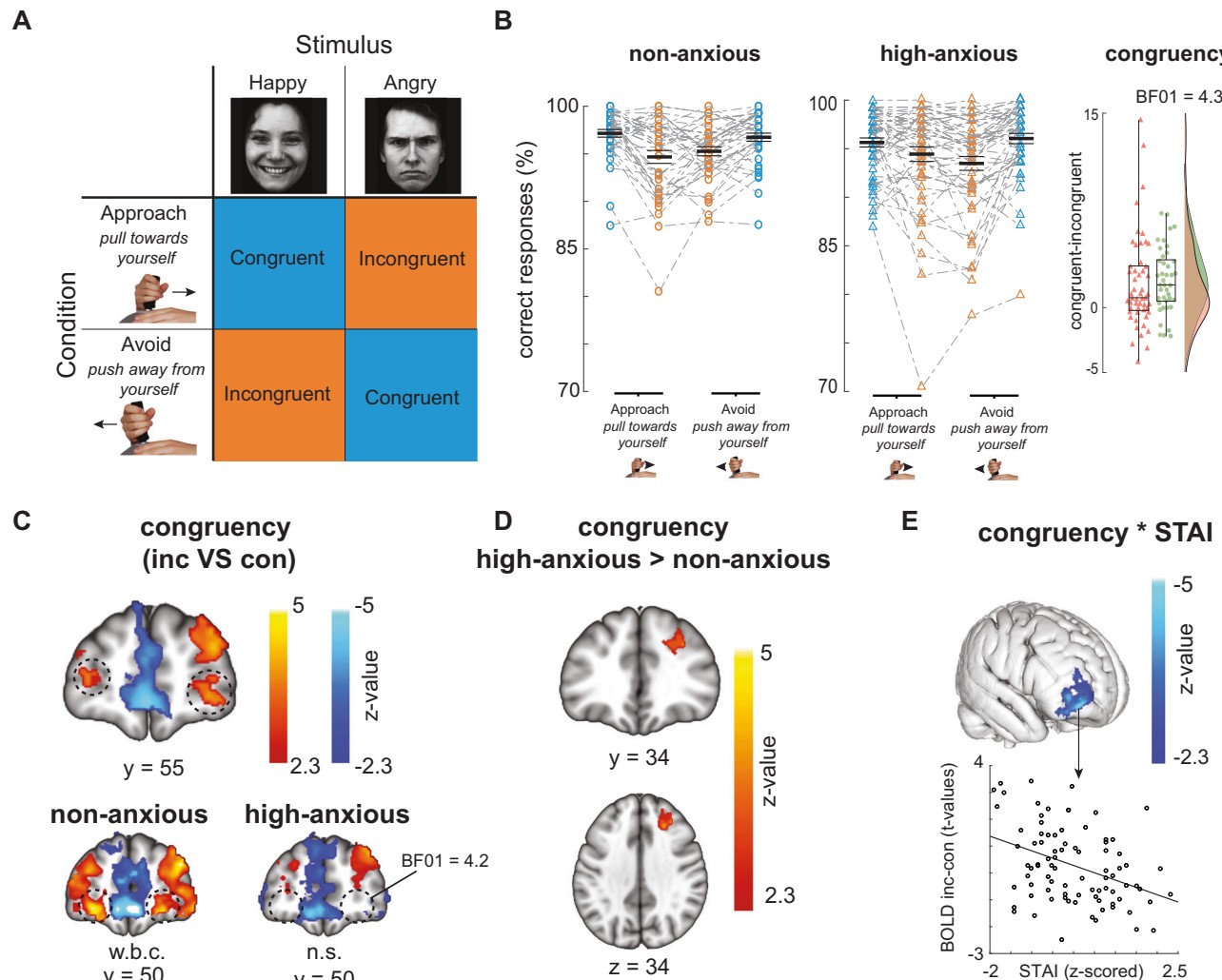

**Fig. 2 | Controlling emotional action tendencies is costly and recruits lateral frontal pole (FPl) differently as a function of anxiety. A** Schematic representation of the Approach-Avoidance task. Participants approach and avoid happy and angry faces by pulling or pushing a joystick towards or away from themselves. Approaching angry and avoiding happy faces requires control over automatic action tendencies that bias towards the opposite. Faces depicted here are "af02has" and "am14ans" taken from the Karolinska Directed Emotional Faces database: https://kdef.se/. **B** Participants make more mistakes in the incongruent as compared to congruent trials. There is no difference in behavioral congruency effect between the non-anxious ($N = 41$; data for this control group already reported in Bramson et al. 2020a) and high-anxiety group ($N = 52$). There is moderate evidence (Bayes Factor ($BF_{01}$) = 4.3) for an absence of effect of group on behavioral congruency, rightmost panel. Boxplots represent mean and 25% percentiles. Lines extend toward maximum values. **C** Controlling emotional action tendencies (incongruent > congruent trials) activates bilateral FPl (in orange; highlighted by dashed black circle; congruent>incongruent in blue). However, the high-anxiety group does not show the neural congruency effect in FPl (bottom panels; w.b.c. whole brain corrected, n.s. non significant). A Bayesian analysis provided moderate evidence for the absence of a neural congruency effect in the FPl of the high-anxiety group ($BF_{01}$ = 4.2). **D** There are differences in neural congruency effects between high-anxious and non-anxious in dorsolateral frontal cortex [24 30 34]. **E** Whole-brain search for a negative correlation between anxiety score and neural congruency effect across both groups showed an effect in the FPl, supporting the suggestion that more anxious participants recruit FPl less when controlling emotional behavior. Scatterplot depicts extracted t-values from this cluster and their relationship with STAI scores for interpretative purposes.

emotional-action control to frontal cortex located dorsolateral and posterior to the FPl used by the non-anxious group. Additional exploration of a parametric relationship between anxiety scores and neural congruency effects supports the notion that trait anxiety reduces reliance on FPl during emotion-control. Namely, participants with higher anxiety scores (across both groups) showed reduced neural congruency effects in FPl (max z = 4.24, p = 0.0004; [40 56 −4], Fig. 2E, corrected for multiple comparisons over the whole brain). Interestingly, across both groups, those participants that recruited FPl the least, relied most on dlPFC, ρ(91) = −0.22, r = 0.038, again suggesting that dlPFC compensates for reduced recruitment of FPl. Behavioral congruency was also differentially related to neural congruency in dlPFC between groups b = 0.11 [0.02 0.19]. Namely, in the high-anxious group, neural congruency effects in dlPFC correlated negatively with behavioral

congruency; ρ(50) = −0.28, p = 0.04, whereas this is not the case for the non-anxious participants, ρ(39) = 0, p = 0.98.

## Anxious participants have a more excitable FPl

Next, we tested whether neurophysiological and structural traits could explain the reduced FPl engagement observed during emotion control in high-anxious participants. We reasoned that the observed between-group differences could reflect reduced functionality of the FPl in the anxious participants. Reduced FPl functionality could emerge from weak responses, i.e. lower excitability and connectivity of this region. Alternatively, this region might have tonically high excitability and strong amygdala input, such that even mild emotional challenges could saturate its neural range[30,31]. Therefore, we used MRS and DWI to measure indices of excitation/inhibition balance and of structural

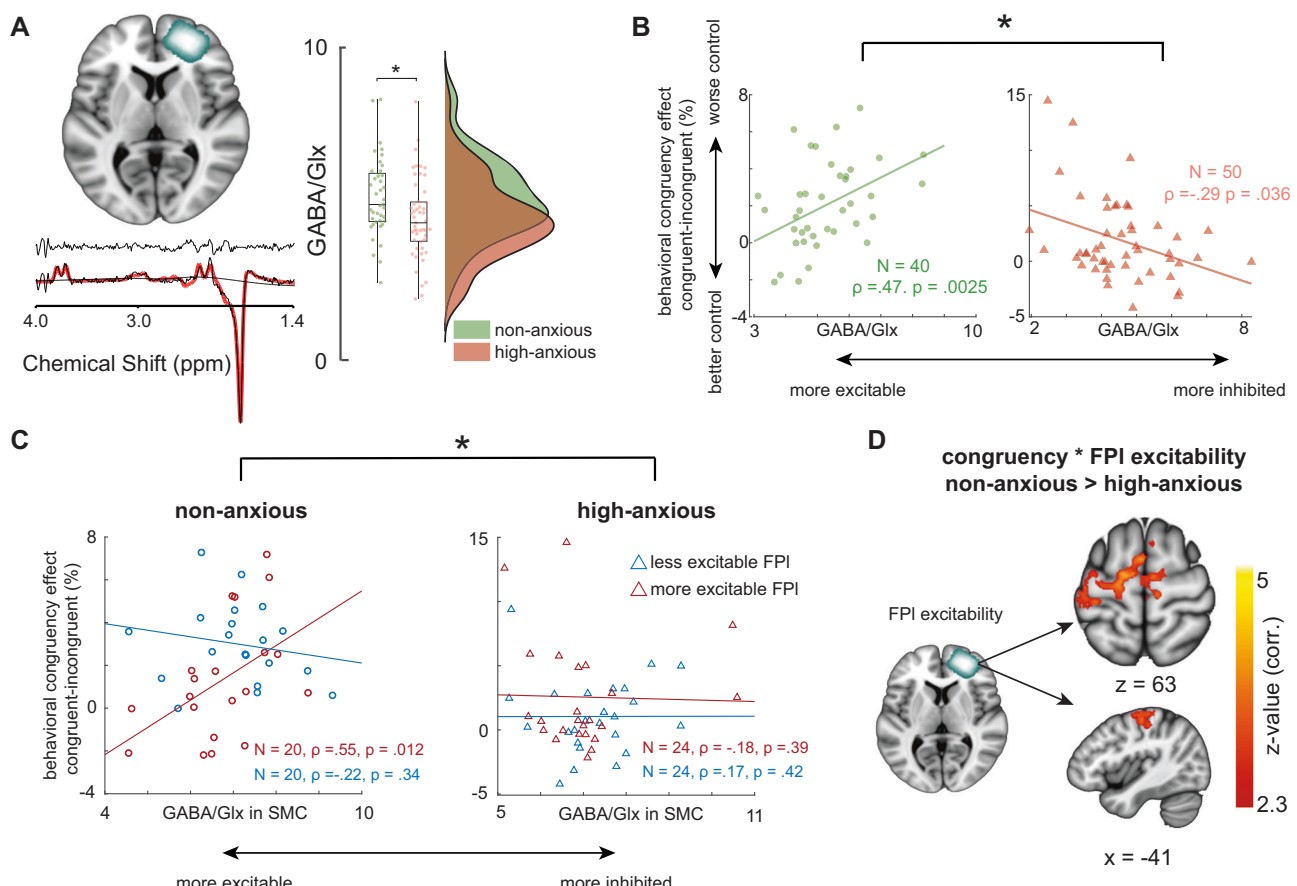

**Fig. 3 | Behavioral and neural indices of emotional-action control differentially depend on lateral frontal pole (FPl) excitability for high-anxiety versus non-anxious participants. A** Average right FPl voxel location quantifying GABA/Glx ratio (top $N = 90$), representative trace of MRS signal residuals (middle, in black), and corresponding MRS spectrum (bottom, in black) with LCmodel fit (in red). Excitability is higher for high-anxious individuals (less GABA relative to Glx) in FPl ($t(88) = 2.3$, $p = 0.02$; two-sided test, asterix represents $p < 0.05$) but not in sensorimotor cortex (SMC) and occipital cortex (supplementary fig. 2). Boxplots represent mean and 25% percentiles. Lines extend toward maximum values. **B** Behavioral congruency effects depend on group and local FPl excitability, $b = 0.19$ CI [0.1 0.28]. In the non-anxious group, participants with more excitable FPl (less GABA vs Glx) show better control over emotional action tendencies (smaller

behavioral congruency effects), $\rho(38) = 0.47$, $p = 0.0025$. In the high-anxious group, this relationship is reversed; $\rho(48) = −0.29$, $p = 0.036$, which does not survive correction for multiple comparisons across the three MRS voxels. **C** For those non-anxious participants that have a relatively excitable FPl the behavioral congruency effects also depend on SMC excitability $b = −0.1$, CI [−0.21 −0.0002]. This is not the case for the high-anxiety group. **D** FPl excitability assessed in a baseline session correlates with neural congruency effects in SMC for non-anxious but not for high-anxious individuals. Combined with the behavioral results, this finding suggests that FPl excitability determine its effectiveness in influencing SMC neural activity during emotional action control. All tests reported here are two-sided. Asterix in 3B & 3 C shows significant differences between conditions (CI does not contain 0 in Bayesian mixed effect models).

connectivity, respectively. We acquired MRS scans from right FPl (Fig. 3A), left SMC and left occipital lobe (Supplementary Fig. 2). The first two locations are known to support the implementation of emotion control[3], and the latter location provides a control region. Left SMC was selected because it has been shown to be under FPl control when selecting affect-incongruent movements with the right hand[3,6]. Within each region, we extracted estimates of GABA and Glx, a proxy for glutamate levels, and calculated an index of neural excitability (GABA/Glx ratio[32]). In FPl, local GABA/Glx ratio was lower (less GABA as compared to Glx; more excitability) in the high-anxiety group as compared to their non-anxious peers, $t(88) = 2.3$, $p = 0.02$, suggesting that the high-anxiety participants had a more excitable FPl (Fig. 3A, right panel). This was not the case in SMC and occipital cortex, both $t < 1.3$, $p > 0.19$, supplementary fig. 2, indicating that the differences in GABA/Glx between groups are anatomically specific to FPl and not a consequence of whole-brain changes in excitability. The GABA/Glx ratio in FPl was also related to behavioral indices of emotion control, differently for the high-anxiety and non-anxious group (three-way interaction between behavioral congruency (congruent vs incongruent), group (high-anxious vs non-anxious), and FPl GABA/Glx ratio, $b = 0.19$ CI [0.1 0.28]). Post-hoc

comparisons (Fig. 3B) show that non-anxious participants with more excitable FPl had better control over emotional action, $\rho(38) = 0.47$, $p = 0.0025$ (Fig. 3B, in green). Importantly, this relationship is reversed in the high-anxiety group $\rho(48) = −0.29$, $p = 0.036$, an indication that high-anxiety participants with more excitable FPl had less control over emotion action (Fig. 3B, in red). Behavioral congruency effects were also related to the GABA/Glx ratio in SMC, but only in non-anxious participants with more excitable FPl (Fig. 3C; four-way interaction between behavioral congruency (congruent vs incongruent) * group (high-anxiety vs non-anxious) * FPl GABA/Glx * SMC GABA/Glx, $b = −0.1$, CI [−0.21 −0.0002]). This interaction complements earlier findings using the same experimental paradigm, showing that FPl interacts with left SMC to implement control over affect-incongruent actions that are performed with the right hand[3,6]. Finally, neural congruency effects in SMC correlate with FPl excitability for the non-anxious group, but not for the high-anxiety group (effect of congruency * group * FPl GABA/Glx ratio in left SMC [−20 −20 66], whole-brain corrected, Fig. 3D). Taken together, these MRS data suggest that anxious participants rely less on previously established long-range FPl influences over SMC for controlling emotional actions[3,27].

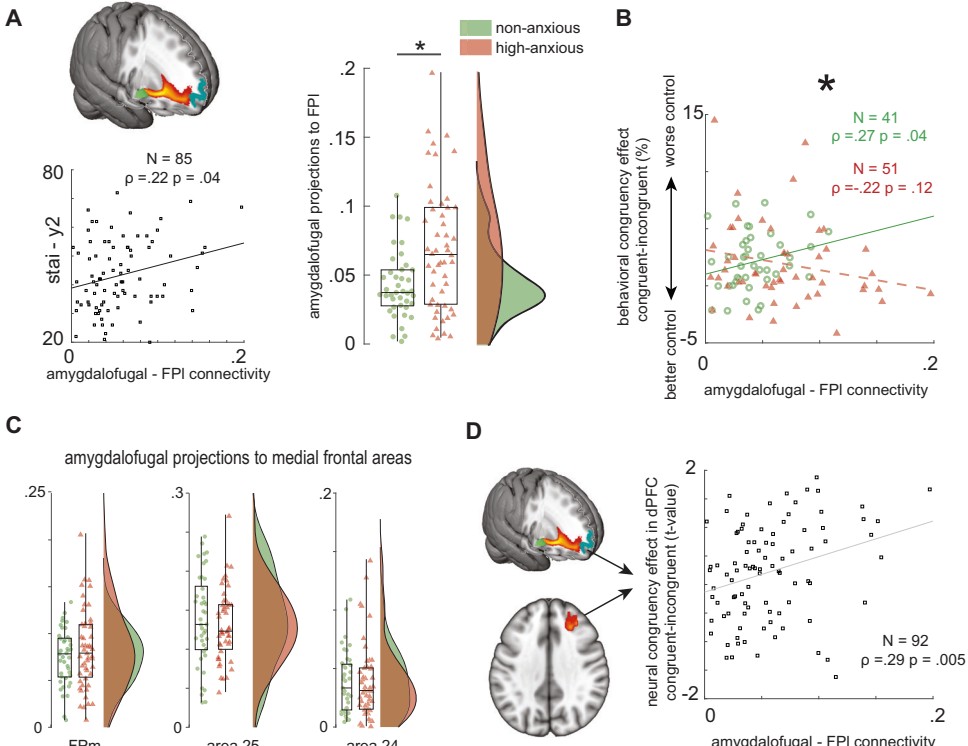

**Fig. 4 | Amygdalofugal projections to lateral frontal pole (FPl) differently support emotion control in non-anxious versus high-anxiety participants.** **A** High-anxious participants (in red) have stronger amygdalofugal projections to as compared to their non-anxious peers (in green); $t(90) = 3.3$ $p = 0.0014$, two-sided test, asterix represents $p < 0.05$. Across the groups, anxiety correlates positively with amygdalofugal-FPl connections. Boxplots represent mean and 25% percentiles. Lines extend toward maximum values. **B** In the high-anxious group, the strength of those projections is not related to behavioral congruency. In contrast, in the non-anxious group, the strength of those projections is related to behavioral congruency effects (reported before in Bramson 2020b). Asterix represents the significant difference between the slopes, $b = 0.14$, CI [0.02 0.26] (CI does not contain zero). **C** Those differences in amygdalofugal projections between non-anxious ($N = 41$) and high-anxious ($N = 51$) are anatomically specific, as they do not extend to other regions innervated by the amygdalofugal bundle, such as medial frontal pole (FPm), area 24 and area 25. Boxplots represent mean and 25% percentiles. **D** Across both groups, participants with stronger amygdalofugal projections to FPl show stronger neural congruency effects in dorsolateral prefrontal cortex $\rho(90) = 0.29$, $p = 0.005$, two-sided test.

To assess whether the results presented above can be attributed specifically to GABA or Glx alone, we repeated the main analyses by considering GABA and Glx independently, as a proportion of Creatine concentration. There was no difference in either FPl GABA/Cr ratio: $t(89) = 1.14$, $p = 0.25$, or FPl Glx/Cr ratio: $t(89) = 0.29$, $p = 0.77$ between groups. There were also no correlations between behavioral congruency effects and FPl GABA/Cr or Glx/Cr ratios; all $\rho < 0.2$, $p > 0.16$. Combined, these results suggest that the ratio between GABA and Glx is important for FPl-based emotional action control, and that it is specifically the ratio between inhibition and excitation in FPl that is different in high-anxious as compared to non-anxious individuals.

### Anxious participants have stronger amygdalofugal projections to FPl

To test whether structural amygdalo-frontal connectivity changes contribute to altered FPl emotional-action control in highly anxious, we acquired DWI. Specifically, we quantified the strength of amygdala projections to the frontal cortex via the amygdalofugal fiber bundle[4]. This measurement is grounded on previous work showing that a substantial portion of variance in behavioral emotional control is accounted by the strength of structural connections from the amygdala to FPl[27]. There are three main findings. First, amygdalofugal bundle projections to the frontal pole were stronger for high-anxious participants than for their non-anxious peers, $t(90) = 3.3$ $p = 0.0014$, Fig. 4A. Second, this between-group difference was anatomically constrained to FPl and area 46 ($t(90) = 3.0$, $p = 0.0027$), since the between-groups difference did not extend to amygdalofugal projections to the medial prefrontal cortex (medial frontal pole; BA 24 and

25) i.e., other regions innervated by the amygdalofugal bundle, (all $(t) < 1.08$, $p > 0.28$; Fig. 4C). Third, the relationship between amygdalofugal anatomy and behavioral indices of emotional control changed between the two groups (three-way interaction between behavioral congruency * group * DWI, $b = 0.14$, CI [0.02 0.26], Fig. 4B). In the non-anxious group, the strength of amygdalofugal projections to FPl showed a positive relationship with the behavioral congruency effect, $\rho(39) = 0.27$ $p = 0.04$ (one-sided test; this group was part of the replication sample in[27]). In contrast, emotional-action control in high-anxious participants was not related to the strength of amygdalofugal projections to FPl, $\rho(49) = -0.22$ $p = 0.12$. These observations are supported by differential correlation between amygdalofugal projection strength and neural congruency effects. Namely, in sensorimotor and medial prefrontal cortex, there are between group differences in the relationship between amygdalofugal-FPl projections and neural congruency effects (supplementary figure 3B). Whereas non-anxious participants show strong correlations between amygdalofugal tract strength to FPl and neural congruency in SMC, high-anxiety participants show correlations between amygdalofugal tract strength and congruency effects in mPFC [0 50 2] and ACC [0 38 22], an effect that is not observed in the non-anxious group, supplementary figure 3B.

### Amygdalofugal-FPl projections predict dorsolateral prefrontal cortex involvement during emotional action control in anxious participants

Finally, because the increased activation in dorsolateral frontal cortex in high-anxiety participants (Fig. 2D) could point to a compensatory mechanism for the observed FPl-related deviations, we tested whether

it was scaled to the magnitude of the increased excitability and amygdalofugal innervation of FPl. There was a positive correlation between dorsolateral prefrontal neural congruency effects (Fig. 2D) and amygdalofugal projections to FPl: $\rho(90) = 0.29$, $p = 0.005$, suggesting that those anxious participants that receive more amygdala projections to FPl may compensate by recruiting dorsolateral frontal cortex, Fig. 4D. There was no relationship between FPl GABA/Glx ratio and strength of neural congruency effects in dorsolateral frontal cortex congruency effects, $\rho(89) = -0.15$, $p = 0.15$, nor was there a direct correlation between amygdalofugal connectivity and FPl excitability $\rho(88) = 0.0$, $p = 0.99$. However, neural congruency effects in those regions that showed a stronger dependency on amygdalofugal connections in the high-anxious group (supplementary fig. 3B) also correlated with neural excitability in FPl, $\rho(88) = -0.27$, $p = 0.0095$: participants with higher excitability in FPl show more compensatory activity in medial and dorsolateral frontal cortices. This again supports the suggestion that these medial and dorsolateral prefrontal regions compensate for FPl dysfunction, either due to increased FPl structural innervation and/or increased excitability.

## Discussion

This study uses neurochemical, structural, and functional measures to characterize neural circuits supporting control of emotional action tendencies in individuals with high-anxiety. There are three main findings. First, anxious individuals use dlPFC, rather than FPl as their non-anxious peers, to implement control over emotional action tendencies. Second, FPl in anxious individuals might receive stronger input from the amygdala via more extensive amygdalofugal pathway connections, and the magnitude of that structural connection predicts the degree of FPl-dlPFC shift during the implementation of emotional control. Third, FPl in anxious individuals is highly excitable, and the excitation/inhibition balance of that region becomes decoupled from behavioral and neural indices of emotional action control, whereas those structure-function relationships are observed in non-anxious individuals. These findings identify a circuit-level neural vulnerability in anxious individuals, and delineate a mechanism explaining how emotional challenges in those individuals could generate a neural bottleneck in the control of emotional action tendencies.

Reduced involvement of FPl in anxious individuals fits with observations of failing FPl recruitment in a number of emotional disorders[26,33,34]. Here, with the combined evidence from neurochemical, structural and functional differences between high-anxious and non-anxious individuals, we add mechanistic insight into that failure. We show reduced involvement of FPl in solving emotional challenges in anxiety, in the face of stronger structural connectivity with amygdala and increased excitability of the same region. Speculatively, the increased FPl excitability in high-anxiety participants could saturate its local circuitry, limiting its ability to fine-tune responses to emotional cues[35] and indiscriminately processing any affective information provided by the stronger amygdala connections. This possibility fits with clinical features observed in anxiety disorders, such as excessive sensitivity to threatening information and overgeneralization of threat information to unrelated situations[36], clinical features that are associated with changes in prefrontal functioning[37-40]. Putatively, the combination of overexcitable FPl and stronger amygdala afferences when controlling emotional actions we observe in high-anxious, might make it difficult for anxious individuals to maintain their private sense of confidence in their opinions when conforming to social norms, a role attributed to FPl[41]. Furthermore, given degeneracy in emotion-related neural circuits[42], the effects of a saturated FPl tuning might alter those circuits, inducing other prefrontal control nodes to take over FPl computational contributions[43-45]. Here, we show that even a mild emotion-regulation challenge results in an anxiety-related shift from FPl to dlPFC, while preserving behavioral performance. However, when anxious individuals face stronger challenges, the same circuit-

level shift might not be able to support adequate emotional control. FPl is able to combine affective information with contextual rules for flexible adaptation of emotion control strategies[2,5,7], whereas dlPFC might not be able to move beyond simple maintenance of task rules[46]. In addition, reliance on dlPFC instead of FPl might create further vulnerabilities for anxious individuals, for instance in situations where dlPFC is taxed with multiple demands, leading to dlPFC hyperactivation when social anxiety participants speak in public[47,48] or manipulate information in working memory[40]. Although we cannot speak to the subjective components of anxiety from our data, it becomes relevant to test how the structure-function relationships we observe relate to individual subjective experiences, given the proposed role of FPl in the consciousness of emotion[49,50].

The finding of increased FPl excitability in high-anxiety participants sharpens the range of possibilities for prevention and treatment interventions. Previous work has shown that emotional-action control in healthy participants can be disrupted by increasing inhibition of FPl by means of transcranial magnetic stimulation[8]. The current findings make it conceivable that this same manipulation might in fact rescue control in anxiety. Patients might also benefit from increasing local inhibitory rhythms in FPl[3,51], for instance by using electrical stimulation tuned to the theta-gamma coupling between FPl and sensorimotor cortex[6,52]. However, it is well possible that overactivation of dorsolateral and medial frontal cortices in anxiety will have to be dampened concurrently, to effectively control the frontal network dynamics and bring FPl-SMC communication within its physiological range[53,54].

The observation that high-anxiety participants show more excitable FPl, combined with anatomically specific increases in innervation from the amygdalofugal bundle, appear relevant for the notion that phylogenetic novelties, like human FPl[20], offer points of vulnerability[19]. Given FPl position in the frontal control hierarchy[43,44], its overexcitability is likely to percolate noise across multiple internal set-points, a situation hard to correct by lower-level controllers[55]. Furthermore, the human-specific amygdalofugal access to FPl, besides allowing for context-dependent emotional control, could also broadcast affective information to lateral prefrontal circuits, particularly so when stronger amygdalofugal input reaches an overexcitable FPl[19,37]. This putative loss of afferent selectivity becomes particularly relevant given the effects of exposure to stress-induced glucocorticoids, namely increased glutamatergic activity and neural excitability in prefrontal cortex following acute stress exposure[56], as well as dendritic loss in prefrontal regions following chronic stress exposure[57,58]. Fittingly, increased glucocorticoid stress-responsiveness as well as stress-induced cortisol have been linked to diminished control over social approach-avoidance tendencies in patients with social anxiety disorder[59,60].

The amygdalofugal bundle contains fibers stemming from both basal and lateral amygdala nuclei as well as the central nucleus. Although it is unclear where the amygdalofugal projections to FPl originate specifically, it is likely that these projections stem from the basal or lateral nucleus, given that most of the central nucleus projections terminate in the Bed Nucleus of the Stria Terminalis[16], and the basal nucleus is considered the main output node of the amygdala complex[61]. It is possible that the projections to FPl are an extension of magnocellular basal nucleus projections that, in macaque, extend to the medial part of BA10 (FPm in humans)[61]. Although connections between agranular regions of the medial prefrontal cortex are dominated by efferent fibers that project toward the amygdala complex, BA10 shows the opposite pattern. Namely, BA10 receives stronger input from the amygdala than vice-versa[24]. It is plausible that the same is true for lateral frontopolar cortex in humans. Our results support recent suggestions that FPl arbitrates between imagined and veridical threat on the basis of magnocellular inputs[62]. Increased amygdala projections might make it difficult for anxious individuals to correctly attribute the assessed dangers projected to FPl to imagined or veridical

threats. FPls potential role as an arbitrator in threat imagery has mostly been described in terms of its potential involvement of intrusive memories in PTSD[62] and more generally fits recent views on the role of the FPl in emotional experience, such as anxiety, and its regulation[49]. Interestingly, in healthy individuals increased FPl activation during emotion control in our task can protect against the development of PTSD symptoms after trauma[9], and exposure therapy has been shown to restore frontopolar function in those PTSD patients that benefit from treatment[63].

The inclusion of only male participants in the non-anxious group is a limitation of this study. However, we consider it unlikely that the neural and behavior congruency effects can be explained solely based on this factor. Namely, both male-only[3,29] and female only[64] studies using the AA task have shown FPl recruitment, and large scale mixed samples did not find differences between males and female participants in FPl engagement[9,65,66]. We also consider it unlikely that males and females differ in specific characteristics of FPl linked to emotional-action control, such as the structural connections from amygdala to FPl and FPl neurochemical profile, in the context of group-matched amygdalofugal projections to medial prefrontal cortices and excitability in SMC and V1 (Fig. 4C & supplementary fig. 2A, B). Although gender differences in the development of GABA concentration across the lifespan have been reported[67], large scale studies comparing male and female participants did not show gender differences in the relationship between GABA and Glx in posterior[68], or prefrontal cortex[69]. Further, although absolute levels of GABA might be different between males and females, the relationship between GABA and glutamate does not seem to vary across gender[70]. Gender differences have been shown in the relationship between anxiety and white-matter connectivity between amygdala and prefrontal cortices[71,72]. However, these were based on whole-bundle average estimates of structural integrity in the Uncinate Fasciculus, rather than differences in relative strength of projections. Microstructural assessment of amygdalofugal white-matter properties do not differ between males and females[73]. Future studies could more stringently test the potential influence of gender differences to amygdalofugal connectivity and FPl neural excitability.

Recent work has pointed out how small sample studies reporting brain-behavior correlations can suffer from inflated effect sizes[74]. Although this is a problem that should be solved through large confirmatory efforts, targeted small-sample studies with high signal-to-noise can provide an hypothesis generating role[75]. Accordingly, in this study, we tried to maximize signal-to-noise within individuals by presenting more than 550 trials per participant over two sessions. Furthermore, our analyses were targeted at a specific neural circuit revolving around FPl, a circuit that has often been associated with emotional action control[2,3,8], including in large prospective samples[9], supporting the potential relevance of the current findings.

The relationship between GABA/Glx ratio and behavioral congruency could not be attributed to effects of GABA (vs creatine) or Glx (vs creatine alone, suggesting emotional action control depends on relative inhibition/excitation in lateral frontal pole, rather than inhibitory or excitatory tone as such. Several reports have shown robust relationshisps between GABA and Glx using 3T[68,76,77], indicating that it can be used as a proxy for E/I ratio. At 3 Tesla, glutamine in the Glx signal could add noise to the GABA/Glx ratio[70], but this effect is controlled for by the experimental design of this study. However, high-field MRS would be required to estimate FPl excitability in high-anxious participants when tailoring individualized treatment interventions.

In summary, we show that, in humans, anxiety is associated with inefficient involvement of FPl during emotional control. We provide evidence for a functional anatomical shift in the implementation of emotional control in anxious individuals, from FPl to dlPFC. This functional anatomical shift is linked to changes in the strength of amygdalofugal projections to FPl and complemented by FPl over-excitability. This shift might explain why highly anxious individuals

struggle to implement flexible emotional action selection during challenging emotional situations, and it suggests interventions to normalize FPl activity in anxiety disorders.

## Methods

### Ethical approval
The study was approved by the local ethics committee (CMO2014/ 288). All participants gave informed consent for participation and for the publication of anonymized data.

### Participants
Fifty-two high-anxious (13 males) and forty-four non-anxious (all males) students of the Radboud University Nijmegen participated in this experiment after giving informed consent. Two non-anxious participants were excluded because they did not attend the whole experiment; one participant was excluded because they failed to comply with the task instructions. All participants had normal or corrected to normal vision and were screened for contra-indications for magnetic resonance imaging. Participants mean age for non-anxious: 23.8 years, SD = 3.4, range 18–34; for high-anxious mean = 25.66, SD = 4.4, range 20–39. The analyses and sample size for the high-anxious were preregistered at: https://osf.io/j9s2z/?view_only= 5510570459694d619adb5dca4019e9fa, announcing two analyses: a brain-stimulation analysis that is not part of the current paper and the planned comparison between the high-anxious and non-anxious group on amygdalofugal projections to FPl and GABA/Glx interactions with behavior reported here). Data from the high-anxious sample have not been reported on previously.

### Data re-use
The non-anxious sample was used as convenience sample to compare the high-anxious participants to because they had been subjected to exactly the same study protocol. fMRI and behavioral analyses in this convenience sample have already been reported in ref. 6. The DTI data and correlation with behavior have been included in ref. 27 as part of the replication sample. These findings therefore do not constitute replications of the effects in the AA task. The MRS data and consequent analyses have not been reported previously. The analyses and sample size of the non-anxious group were previously preregistered: https:// osf.io/m9bv7/?view_only=18d58e2351b14584b6e688599472534e analyses based on which we decided to recruit a high-anxious group for comparison.

### Procedure
The data reported in this manuscript were acquired over three different days as part of two brain stimulation studies. On the first day, we acquired a structural T1 scan, immediately followed by three magnetic resonance spectroscopy scans over FPl, SMC, and occipital cortex (in a fixed order). In two participants, time constraints prevented the acquisition of MRS over occipital cortex. On the second and third day, participants practiced the approach-avoidance task for five minutes (second day only), before starting a 35 min task performance whilst fMRI was acquired. During task performance, the participants received transcranial alternating current stimulation (tACS) over FPl and SMC (reported in ref. 6). Analyses on the different stimulation protocols are described in ref. 6. For the purpose of this report, we combined fMRI and behavioral data collected across the three stimulation conditions.

### Emotional Approach-Avoidance (AA) task
During the two fMRI sessions, participants performed a social-emotional approach-avoidance task designed to study control over social-emotional action tendencies[8,27]. All stimulus material was presented using Presentation software version 16.4 (https://www.neurobs. com/). In the AA task, emotional faces were presented on screen (100 ms). Participants had 2000 ms to respond. In the "congruent"

trials, participants were instructed to pull the joystick towards themselves as fast as possible when they saw a happy face, and push it away from themselves when they saw an angry face. These trials are congruent with the automatic action-tendencies to approach happy- and avoid angry faces[78,79]. In the incongruent trials, participants were asked to push the joystick away when they saw a happy face, and pull it towards themselves when they saw an angry face. These trials are incongruent with the participants automatic action tendencies. Overriding automatic action tendencies requires a complex form of cognitive control that operates on the interaction between emotional percepts and the emotional valence of the action[8,79,80], neurally implemented through FPl control over downstream regions[6,8,27]. Participants responded using a joystick that could move only along the participant's midsagittal plane. Written instructions were presented on the screen for a minimum of 30 s prior to the start of each block of 12 trials. The terms "congruency" and "approach" or "avoid" were not mentioned to the participants. Congruent and incongruent conditions alternated between blocks. Trials started with a fixation cross presented in the center of the screen for 500 ms, followed by the presentation of a face for 100 ms. Participants were asked to respond as fast as possible, with a maximum response time of 2000 ms. Movements exceeding 30% of the potential movement range of the joystick were taken as valid responses. Online feedback ("you did not move the joystick far enough") was provided on screen if response time exceeded 2000 ms. Each participant performed 288 trials on each of the two testing days, yielding 576 trials in total, equally divided between congruent and incongruent conditions.

## Materials and apparatus
All magnetic resonance images were acquired using a 3 T MAGNETROM Prisma MR scanner (Siemens AG, Healthcare Sector, Erlangen, Germany) using a 32-channel headcoil for the structural T1 and MRS scans, and a 64-channel headcoil for the functional images.

High-resolution anatomical images were acquired with a single-shot MPRAGE sequence with an acceleration factor of 2 (GRAPPA method), a TR of 2400 ms, TE 2.13 ms. Effective voxel size was $1 \times 1 \times 1$ mm with 176 sagittal slices, distance factor 50%, flip angle 8°, orientation A ≫ P, FoV 256 mm.

Magnetic resonance images were acquired using a MEGA-PRESS WIP sequence (SIEMENS) with TE = 68 ms, TR = 1500 ms, water suppression at 4.7 ppm (CHESS[81] and acquisition bandwidth of 1200 Hz. In one of every two acquisitions, a refocusing pulse was applied at 1.9 ppm. Subtracting these signals from the non-refocused scans showed GABA resonance at 3.00 ppm. As a proxy for Glutamate levels we used Glx, which consists of combined Glutamate and Glutamine levels. Glx was estimated from unedited spectra following earlier protocols[82]. MRS measurements were acquired from right FPl[20,27], and from left SMC[3]. A voxel in the right occipital cortex was also measured as a non-task-related control. Voxel size was $1.8 \times 1.8 \times 2.5$ cm[82].

The field of view of the functional scans acquired in the MR-sessions was aligned to a built-in brain-atlas to ensure a consistent MR field of view across days. Approximately 1800 functional images were continuously acquired in each scanning day using a multi-band 6 sequence, 2*2*2 mm voxel size, TR/TE = 1000/34 ms, Flip angle = 60°, phase angle P ≫ A, including 10 volumes with reversed phase encoding (A ≫ P) to correct image distortions.

Diffusion-weighted images were acquired using echo-planar imaging with multiband acceleration factor of 2 (GRAPPA method), multiband acceleration factor = 3. We acquired 93 1.6 mm thick transversal slices with voxel size of $1.6 \times 1.6 \times 1.6$ mm, phase encoding direction A ≫ P, FoV 211 mm, TR = 3350, TE = 71.20. 256 isotropically distributed directions were acquired using a b-value of 2500 s/mm². An additional volume without diffusion weighting with reverse phase encoding (P ≫ A) was also acquired.

Emotional faces used in the AA task were taken from open source databases[83–85] and adapted for use in this task. Full list of used image identifiers is provided in the supplementary materials.

## Analyses – MRS
Spectroscopy data were analyzed using LCmodel software[86]. After frequency alignment, eddy current correction, phase- and baseline corrections, the relative concentrations of neurotransmitters were estimated using basis sets, against which we fitted the acquired signals in both the edited and non-edited spectra. Spectra quality control was based on several estimates of signal quality; the % SD provided by LCmodel, which reflects the Cramer-Rao lower bound; full-width half maximum FWHM, estimates of signal to noise ratio provided by LCmodel and visual inspection. Inhibitory tone was then calculated by calculating the ratio between GABA and GLx.

## MRS data quality assessment
MRS signal quality was assessed based on a combination of visual inspection, Cramer-Rao lower bound (CR; cutoff < 30), signal-to-noise ratio (snr; >10) and full-width half maximum provided by LCmodel. Signal quality was good in FPl, and very good in SMC and occipital cortex[87]. Based on signal quality from FPl we excluded one participant from the non-anxious group (CR = 46, snr = 6) and two participants from the high-anxious group (CR > 36). All analyses of FPl and SMC data were performed on the remaining participants. Control analyses using GABA/Glx ratio extracted from occipital cortex contained 38 participants in the control group and 50 in the high anxiety group. For this region, four participants were not measured due to time constraints, one was removed based on bad quality data. Overview of the MRS data quality is presented in Table 1, example MRS spectrum for FPl is depicted in Fig. 3 and example spectra for all regions are depicted in supplementary figure 2A.

## Analyses – behavioral responses
For all behavior analyses we focused primarily on differential error rates between the congruent and incongruent condition, because those have been most strongly linked to both individual differences in structural and functional properties of the FPl system under investigation[6,8,27]. Reaction time analyses are reported in the supplementary materials. Those behavioral metrics were related to the GABA/Glx ratio extracted from FPl, SMC, and occipital MRS-voxels using Bayesian mixed effects models and Spearman's correlation coefficient. Follow-up analyses also considered the strength of amygdalofugal connections to FPl. Bayesian mixed effects models were implemented in R 3.5.3 using the *brms* package[88]. We considered two factors; Group (high-anxious versus non-anxious) and Emotion control (congruent versus incongruent). Follow-up analyses of behavioral performance considered models that also included GABA/Glx estimates or amygdalofugal connection strength to FPl. Given that previous studies have shown that FPl is involved in implementing control over emotional action tendencies[27–29,65] and that it does so by interacting with SMC[3,6] we first set out to assess whether the ability to control emotional action tendencies depends on local FPl and/or SMC inhibitory tone. To test for regional specificity, we also implemented a model regressing behavioral congruency effects against GABA/Glx estimates acquired from the occipital cortex. All models included random intercept for participants and random slopes for the behavioral congruency effect. This model adheres to the maximal random effects structure[89]. Outputs of these models are log odds with credible intervals ("b"). In these analyses an effect is seen as statistically significant if the credible interval does not contain zero with 95% certainty. Significant interactions were further characterized by using Pearson correlation coefficient with Bonferroni correction over the three regions (MRS analyses).

**Table 1 | Magnetic Resonance Spectroscopy data quality assessment**

| Region | GABA | | | Glx | |
|---|---|---|---|---|---|
| | Fwhm (std) | snr (std) | CR (std) | Sd (std) | |
| FPl | 0.059 (0.01) | 15.6 (5.4) | 18.6 (4.7) | 7.18 (1.5) | non-anxious group: 1 excluded (Sd = 46,sn = 6) |
| | 0.061 (0.01) | 16.7 (3.6) | 19.34 (5.49) | 6.4 (1.1) | high-anxious group: 2 excluded (Sd = 68 & 36) |
| SMC | 0.05 (0.01) | 26.1 (4.1) | 13.4 (2.4) | 6.49 (0.87) | non-anxious group: all included |
| | 0.046 (.009) | 26.4 (3.5) | 12.9 (1.7) | 6.65 (1.3) | high-anxious group: 1 excluded |
| Visual | 0.05 (0.005) | 26.3 (3.2) | 13.7 (2.8) | 7.5 (1.46) | non-anxious group: 1 excluded, 2 missing |
| | 0.05 (0.005) | 25.5 (3.5) | 15.4 (3.9) | 7.6 (1.22) | high-anxiety group: 2 missing |

This table shows GABA and Glx data quality measures for all three voxels; Cramer-Rao lower bound (Sd); Full width half maximum of the spectrum (FWHM); and estimated signal to noise (Snr) for the GABA peak extracted from the edited-spectrum, all provided by LCmodel. We excluded 3 FPl spectra based on a combination of Cramer-Rao lower bound, low signal-to-noise and visual inspection[87].

### Analyses fMRI – preprocessing

fMRI images were analysed using FSL 6.0.0 (https://fsl.fmrib.ox.ac.uk). Images were motion corrected using MCFLIRT[90], and distortions in the magnetic field were corrected using TOPUP[91]. Functional images were rigid-body registered to the brain extracted structural image using FLIRT. Registration to MNI 2 mm standard space used the nonlinear registration tool FNIRT. Images were spatially smoothed using a Gaussian 5 mm kernel and high pass filtered with a cut-off estimated on the task structure. Independent component analysis was run with a pre-specified maximum of 100 components[92]; these components were manually inspected to remove potential sources of noise.

### Analyses fMRI – GLM

First and second level GLM analyses were performed using FEAT 6.00 implemented in FSL 6.0.0. The first-level model consisted of twelve task regressors: Approach angry, approach happy, avoid angry and avoid happy trials were modelled separately for each of the three stimulation conditions (for details on stimulation conditions see Bramson et al. 2020a). In each regressor, each event covered the time interval from presentation of a face until the corresponding onset of the joystick movement. Estimated head translations/rotations during scanning (six regressors), temporal derivatives of those translations/rotations (six regressors), and MR-signals in white matter and cerebrospinal fluid (2 regressors) were included to the GLM as nuisance covariates. Emotional control effects were estimated by comparing incongruent trials (approach angry and avoid happy) to congruent trials. First level models of the two separate sessions were combined using Fixed Effects analyses implemented in FEAT.

Whole brain group effects and their relationship to GABA/Glx ratio's in FPl and SMC, and amygdalofugal connections to FPl were assessed using FLAME 1 with outlier de-weighting[93], making family-wise error corrected cluster-level inferences using a cluster-forming threshold of $z > 2.3$. This threshold provides a false error rate of around 5% when using FSL's FLAME 1[94]. In both whole brain and ROI analyses we used standardized GABA/Glx ratio's extracted from FPl and SMC as regressors. GABA/Glx ratio extracted from occipital cortex was used in a separate control analysis.

### Analyses – diffusion

Diffusion data was preprocessed using FSL FDT 3.0 (https://fsl.fmrib.ox.ac.uk). Susceptibility artefacts were estimated using TOPUP using additional $b = 0$ volumes with reverse phase coding direction[91]. Data were then corrected for potential distortions during eddy currents and movement by using the EDDY tool[95]. Crossing fibers were estimated using BedpostX with default settings[96].

We reconstructed the amygdalofugal pathway using FSL's Prob-trackX tool using waypoint and exclusion masks by[4]. In brief, a seed was placed in the white matter punctuating the extended amygdala and substantia innominata: MNI: [−7 3 −9]. Tracking was constrained by using an all-coronal waypoint mask at $y = 22$. Tractography was further constrained to exclude CSF and across hemisphere connections (which are not part of the amygdalofugal pathway), and not

allowed to extend caudally from the seed location, up to the $y = 25$ coronal plane. Connection strength was normalized and log transformed within each participant. Next we extracted the total amount of times the tractography entered the FPl based on the white-matter border masks provided by[20]. These values were compared between groups and used as regressors in behavioral and neural congruency analyses on reaction time and percentage correct using Spearman's correlation coefficient.

### Analyses – statistics

Statistical models testing behavioral congruency effects across and between groups, and derived models adding covariates were run in a step-wise fashion. We first compared correct responses between congruent and incongruent conditions between the different groups: Group (non-anxious vs high-anxious) * congruency (congruent vs incongruent). We then extended this model in two iterations by adding estimates of excitability in a 4-way interaction: Group*congruency* FPl GABA/Glx * SMC GABA/Glx. Amygdalofugal tract strength was added in a separate three-way interaction: Group*Congruency*amygdalofugal tract strength. Significant interactions were assessed by interpreting lower-level interactions resulting from these same models, or post-hoc Spearman correlations. Full models and results for the most important interactions are presented in supplementary table 4. Correlations between neural excitability and behavioral congruency for the different groups were Bonferroni corrected for the three regions of interest. Analyses on functional MRI effects were cluster corrected using a cluster-threshold forming threshold of $z > 2.3$ controlling either for all voxels in the brain (whole brain analyses) or all voxels in the frontal lobe. Spearman correlations were calculated in matlab2020b (www.mathworks.com).

We ran several additional models including a six-way interaction model containing all major parameters of interest to explain behavioral congruency. The model explained correct responses based on congruency(congruent vs incongruent) * Group (non-anxious vs high-anxious) * FPl BOLD congruency * dlPFC BOLD congruency * FPl GABA/Glx * amygdalofugal-FPl tract strength. This approach yielded interactions between congruency*Group*FPl GABA/Glx and Congruency*Group*dlPFC BOLD. Interestingly, splitting up this model did give significant interactions between congruency*group*FPl BOLD, suggesting that estimates of FPl GABA/Glx, FPl BOLD and amygdalofugal-FPl connectivity explain partly overlapping variance between participants.

### Reporting summary

Further information on research design is available in the Nature Portfolio Reporting Summary linked to this article.

## Data availability

This paper is accompanied by source data. The data generated in this study have been deposited in the Donders data repository (data.donders.ru.nl) under the access code di.dccn.DSC_3023010.01_497, with https://doi.org/10.34973/29k2-7p09.

## Code availability

The code generated for this study have been deposited in the Donders repository (data.donders.ru.nl) under the access code di.dccn.DSC_3023010.01_497, with https://doi.org/10.34973/29k2-7p09, or are available upon request to the corresponding author.

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

## Acknowledgements

This work was funded by consolidator grant DARE2APPROACH from the European Research Council (ERC_CoG_772337) awarded to KR and also supporting BB; by a consortium grant from the Dutch Research Council INTENSE (NWO_Crossover_17619) and by (NWO_SSH_406.20.GO.020) awarded to IT and KR. The authors would like to thank Soha Farboud for assistance with visualizations and Siemens Healthcare Nederland B.V. for providing the MEGA-PRESS sequence.

## Author contributions

BB, IT & KR designed the study; BB and SM acquired the data; BB, SM, AN, IT & KR performed the analyses. BB wrote the first draft. BB, SM, IT & KR wrote revised drafts. AN provided feedback.

## Competing interests

The authors declare no competing interests.
