## [Peer Review File · Nature Communications]

Reviewers' Comments:

Reviewer #1:

Remarks to the Author:

This is a timely and important study. The findings are fascinating and add a new way to think about anxiety. I have a few suggestions, which, if responded, would, I think, improve the paper, and would allow me to support publication.

One concern is the relation between the frontal pole and amygdala. The authors write:

...the FPI has access to both medial and lateral cortical circuits through its extensive connections with other frontal, parietal and temporal association areas. In addition, FPI has direct access to information coming from the amygdala via the amygdalo-fugal bundle^{4,20}. In contrast, macaque's amygdalae projects to medial but not lateral prefrontal regions^{24,25}. Accordingly, recent work has suggested that FPI is involved in selecting emotional actions when different alternative options are available⁵.

The above text needs clarification. The implication is that the lateral frontal pole gets amygdala inputs via the amygdalo-fugal bundle. But the next sentence says that in monkeys the amygdala projects to the medial but not lateral frontal pole. This raises several anatomical questions. First, how do we extrapolate from the monkey to the human lateral frontal pole? Although the monkey is our best source of cortical connectivity, the lateral pole of the monkey is considerably more primitive anatomically than humans. Second, the medial frontal pole, which is part of granular PFC is often thrown together with a/dysgranular medial PFC areas, especially in relation to the default mode network. I would suggest carefully evaluating the studies cited showing amygdala connections to granular medial frontal pole to rule out the possibility that that the findings are truly about granular medial frontal pole as opposed to adjacent a/dysgranular medial PFC. Third, and related, there seems to be a contradiction about the amygdala and lateral FP. Which is correct: (1) there are no inputs from amygdala to lateral FP, or (2) the amygdalo-fugal bundle carries amygdala inputs to the lateral FP? I would suggest exploring the literature to determine the purity of amygdalo-fugal bundle. Does it have fibers from other areas besides the amygdala?

If all of the above is clarified, I would want to know more about which regions of the amygdala connect to lateral and medial FP. In general, the basolateral areas of the amygdala connect with neocortex and the central and medial regions connect with mesocortical, a/dysgranular medial cortex. Lateral and medial FP, being part of granular neocortex, might be expected to be connected with the basolateral amygdala. Knowing these details will be very important in understanding if and how amygdala connections with polar PFC contribute to anxiety.

My other concern is about what anxiety itself is. For the anxious person, much of what it means to be anxious centers on how the patient feels subjectively. The only hint of concern about the feeling of anxiety in the paper is the citation of Glassman 2014 study on public speaking. I think the paper could be strengthened by including relevant citations about the FP and conscious experience in general and conscious fear and anxiety. Some suggestions follow.

Bang D, Ershadmanesh S, Nili H, Fleming SM. Private-public mappings in human prefrontal cortex. *Elife*. 2020 Jul 23;9:e56477. doi: 10.7554/eLife.56477. PMID: 32701449; PMCID: PMC7377905.

Brown R, Lau H, LeDoux JE. Understanding the Higher-Order Approach to Consciousness. *Trends Cogn Sci*. 2019 Sep;23(9):754-768. doi: 10.1016/j.tics.2019.06.009. Epub 2019 Jul 30. PMID: 31375408.

Cushing CA, Dawes AJ, Hofmann SG, Lau H, LeDoux JE, Taschereau-Dumouchel V. A generative adversarial model of intrusive imagery in the human brain. *PNAS Nexus*. 2023 Jan 23;2(1):pgac265. doi: 10.1093/pnasnexus/pgac265. PMID: 36733294; PMCID: PMC9887942.

Lau H, Rosenthal D. Empirical support for higher-order theories of conscious awareness. *Trends Cogn Sci*. 2011 Aug;15(8):365-73. doi: 10.1016/j.tics.2011.05.009. Epub 2011 Jul 6. PMID: 21737339.

LeDoux JE. What emotions might be like in other animals. *Curr Biol*. 2021 Jul 12;31(13):R824-R829. doi: 10.1016/j.cub.2021.05.005. PMID: 34256909.

Taschereau-Dumouchel V, Michel M, Lau H, Hofmann SG, LeDoux JE. Putting the "mental" back in "mental disorders": a perspective from research on fear and anxiety. *Mol Psychiatry*. 2022 Mar;27(3):1322-1330. doi: 10.1038/s41380-021-01395-5. Epub 2022 Jan 26. PMID: 35079126; PMCID: PMC9095479.

Reviewer #2:

Remarks to the Author:

I am reviewing the paper "Anxious individuals shift emotion control from lateral frontal pole to dorsal prefrontal cortex; functional, structural and neurochemical evidence" by Bramson et al. In this study, the authors examine lateral frontopolar (FPI) function in control of emotional action using a well validated task and a multi-modal imaging approach, including functional neuroimaging, structural connectivity (DWI), and spectroscopy metrics that reflect the baseline excitability (GABA/Glu ratio) of FPI and PFC regions of interest. The authors contrast FPI & (DL)PFC activation profiles between high-anxious vs. non-anxious individuals, and find that (1) Anxious participants show less "emotion incongruency" activation in FPI compared to non-anxious; (2) Anxious participants show instead increased congruency-sensitive activation in dorsolateral PFC (DLPFC) compared to non-anxious individuals; (3) The magnitude of functional engagement of FPI to the incongruency contrast is inversely correlated with anxiety scores, further corroborating finding (#1). Moreover, the authors attempt to relate these distinct profiles of FPI engagement in their task to other trait-like variation in the structure and excitability of FPI. Here, they find that (4) FPI is more excitable in anxious vs. non-anxious individuals; and the magnitude of FPI excitability is differentially associated with the (behavioral) congruency effect in their task as a function of anxiety (such that non-anxious individuals have better emotional control with more excitable FPI; but vice versa for anxious folks). There are additional findings here that were not readily intuitive to me, which included that (5) the behavioral congruency effect was related to GABA/Glu in SMC in the non-anxious group, and that (6) the neural congruency effect in SMC correlated with FPI excitability only in the non-anxious group. When examining FPI structure, the authors found that (7) amygdalofugal pathway/projections to FPI were stronger in high-anxious individuals and that this group difference was spatially specific to FPI and not amygdalofugal projections to medial PFC (BA24/25); and (8) that the association between amygdalofugal anatomy and emotional control changed between the groups, such that in the non-anxious group only, higher structural connectivity was associated with greater behavioral congruency effects (worse control). They also note that (9) high anxiety participants show associations between amygdalofugal strength and neural congruency effect in mPFC and ACC. Finally, the authors find that amygdalofugal-FPI strength was associated with greater DLPFC, further corroborating finding (2).

On one hand, the quest of determining regional specificity in the PFC for the cognitive control of emotion is timely and in critical need of new data and careful analysis aimed at that goal. The authors have amassed an impressive amount of data to that end and have prioritized intra-subject reliability (collecting a large number of trials per subject; $n > 500$), which is commendable and rarely done in affective neuroscience. Moreover, they use a task that is well validated by the past decade of their work, a body of work that had raised important questions regarding FPI "helping" or "hindering" regulation, which the present manuscript helps to answer. On the other hand, there were aspects of the current manuscript that I found challenging to parse—conceptually, and I also had a few methodological concerns, which if addressed, could strengthen an already impressive body of work.

Major concerns

1. For finding in Figure 2C, I appreciate using Bayesian statistics, but another strong and easily interpretable test is whether there is a significant interaction between Group and the Congruency effect. Same question for Figure 2D: in other words, is FPI (incongruent) significantly more engaged for Non-anxious vs. Anxious, and DLPFC significantly more engaged for Anx vs. Non-Anx?

2. An additional (potentially more direct) test of the authors' interesting conjecture regarding DLPFC vs. FPI engagement in the High Anx group is that the magnitude of Inc>Cong in DLPFC should be inversely correlated with the magnitude of Inc>Cong FPI, either in High Anx alone, or potentially across both groups. In other words, if DLPFC engagement is reflecting some sort of compensatory effect, then it should be inversely related to how much FPI is 'up to the task'. Is that the case? And/or is the delta DLPFC-FPI across subjects related to anxiety and/or the emotional congruency effect?

3. The analysis that included Left SMC (Figures 3C & 3D) (also, why Left and not Right?) did not seem theoretically motivated until I got deep into the Methods section. If the authors consider that set of findings of equal import to the manuscript (vs. being in Supplementary) compared to FPI/DLPFC findings, then additional setting up of the theoretical framework—including prior findings—and stating the hypotheses in the Introduction would be warranted. As is, I found those findings quite difficult to follow and to integrate with FPI and DLPFC results.

4. I found the reverse association between Gaba/Glu and the behavioral congruency effect intriguing (Fig 3B). It suggests there's a point at which FPI function/engagement is 'counter-productive' for Anx folks. I would have liked to see those findings more directly connected to DLPFC function (Fig 2B)—are these two separate stories—i.e. are these effects impacting different individuals? Or are the High Anx individuals with more excitable FPI (less helpful) also the ones recruiting DLPFC more during the task (Inc> Cong)? Can you enter those IVs (GABA/Glu, DLPFC engagement, FPI engagement, amygdalofugal pathway-FPI) in the same simultaneous regression model predicting Anx symptoms or the behavioral congruency effect to ascertain whether these neural metrics are explaining overlapping variance, or possibly exerting 'suppression' (in a regression framework) effects?

(On suppression, see: <https://www.ncbi.nlm.nih.gov/pmc/articles/PMC3780394/>)

5. The gender imbalance across the samples raises questions about how to best address this confound. It was not clear to me exactly how Figure 4B or Suppl. Figure 3 addressed it (though they're mentioned in the Discussion). Is there an additional control sample of non-anxious individuals that includes females that the authors could use for at least a subset of these analysis? (e.g. structural); or vice versa (i.e. a high anxiety group with male subjects only). The uncinate fasciculus (for instance, another important amygdala pathway) is known to show gender differences: <https://ajp.psychiatryonline.org/doi/full/10.1176/appi.ajp.2018.18040425>
<https://www.frontiersin.org/articles/10.3389/fnsys.2016.00093/full>

If no additional (balanced) samples available, can the authors control for gender in their model to see if results hold?

6. Supplementary Figure 3 is quite novel and important; I was surprised to see it in Supplementary. In particular, I would be interested in knowing whether there is an interaction between region (Fpl vs. BA25 and 24) and Group (as for BA24/25 the association with Anx seems numerically reversed). It is also interesting that the amygdalofugal_FPI pathway correlates positively with anxiety, which warrants additional discussion.

7. Figure 4D is very helpful for connecting and interpreting the structural-functional findings across the paper; but here I found Figure 4C difficult to integrate with the rest. Is there spatial overlap between 4C and 4D? How does 4C (medial PFC, also dorsal PFC, interaction with Group and amygdalofugal pathway) help with the interpretation of 4D (greater DLPFC engagement across groups for individuals with greater amygdalofugal pathway to FPI)?

Minor points

1. Intro Line 77: 'deviant recruitment'- awkward phrasing
2. Intro Line 95/Discussion Line 282: I am not sure the present data permit strong inferences regarding FPI afferents vs efferents; can the authors elaborate on this?
3. Is there spatial overlap between the results shown in 2E and in 2C? If so, that would be worth indicating.
4. I believe there is an error when referring to Figure 4; there is no 4E and I think the lettering is off by one throughout.
5. I do wonder whether the authors examined differential amygdala activation profiles across these Anx vs. Non Anx groups and whether those relate to FPI structural, neurochemical, or functional profiles?
6. This is at surface minor but it is important for interpretation. At times the 'congruence' effects seemed to refer to FPI's increased engagement to incongruent vs. congruent, but later it was referred to primarily as the facilitation of action by emotion. Keeping the labeling, terminology (and contrast direction) consistent throughout the manuscript would facilitate parsing through (and connecting) the present results.

Reviewer #3:

Remarks to the Author:

Bramson et al. investigate the role of the brain circuit involving the lateral frontolpolar cortex (FPI), posterior parietal cortex, sensorimotor cortex and amygdala in emotion-action control in subjects with anxiety. They used magnetic resonance spectroscopy, diffusion weighted imaging and functional MRI to target FPI GABA/Glx levels, functional activation of FPI during a mild emotional challenge requiring emotional action control in high-anxious (n=52, 14 male) and non-anxious (n=41, all male) participants. Their main findings are the following: relative to non-anxious subjects, high-anxious participants use more dPFC in emotion-action control, have lower FPI GABA/Glx, and show stronger connectivity in amygdalofugal projection to FPI. They conclude that there is a functional-anatomical shift in emotion-action control in anxious subjects, and this shift is associated with the change in structural connectivity of amygdalofugal projections to FPI and overexcitability in FPI.

Overall, I find that it is very nice to use a multimodal approach addressing the question from functional, structural and neurochemical aspects, which may provide mechanistic insights. I have a few major comments related to the study methodologies and design, which may affect the results, interpretation and conclusion.

Major comments:

1. A "statistical analysis" section is missing in the method part. For example, a lot of 3 or 4 way interaction analysis were done also with some post-hoc analysis. But it is not clear whether the interaction is significant or not. If not, then the post-hoc analysis may be not valid. Was multiple comparison correction applied? Which kind of correction was used? A summary table with statistical results will be useful to have an overview of the results.
2. For neurochemical data measured by MRS, authors used GABA/Glx ratio to evaluate the excitability in FPI. The increased excitability is concluded from reduced GABA/Glx. In fact, reduced GABA/Glx can be led by increases in Glx or decreases in GABA, or decreases in both but with more extent in GABA. Therefore, authors should quantify GABA and Glx separately, which will facilitate the interpretation of the results. Did authors acquire unsuppressed water spectra? You can quantify GABA and Glx separately, instead of using GABA/Glx. In this case, one can know if the change observed is driven by e.g. reductions in GABA or just increases in Glx.
3. Most of the mechanistic interpretation in the discussion is established on increased excitability in the FPI of anxious subjects. As commented above, these discussion part should be revised after looking into individual GABA and Glx results.
4. Another big issue is that the gender is largely biased between high-anxious (only 14 males) and

non-anxious (all males) groups. The authors mentioned that gender has minor effect on the functional FPI engagement. They also claim that the structural connections and FPI neurochemical profile are unlikely affected by gender, which is lack of supporting evidences. At least, the gender effect on brain metabolites has been reported. If gender does have an effect on GABA and Glx, then authors should be careful that the observed difference may be dominated by the gender difference.

Minor comments:

1. Why left SMC, and right FPI and occipital lobe were investigated? Why not stay with the same side?
2. Please specify the snr in MRS data, is it the snr of edited spectra (referring to GABA peak? Or Glx peak?) or non-edited spectra?
3. What is the specific parameter measured from DWI for "connectivity"?

Reviewer #1 (Remarks to the Author):

This is a timely and important study. The findings are fascinating and add a new way to think about anxiety. I have a few suggestions, which, if responded, would, I think, improve the paper, and would allow me to support publication.

One concern is the relation between the frontal pole and amygdala. The authors write:

“...the FPI has access to both medial and lateral cortical circuits through its extensive connections with other frontal, parietal and temporal association areas. In addition, FPI has direct access to information coming from the amygdala via the amygdalo-fugal bundle^{4,20}. In contrast, macaque’s amygdalae projects to medial but not lateral prefrontal regions^{24,25}. Accordingly, recent work has suggested that FPI is involved in selecting emotional actions when different alternative options are available.”

The above text needs clarification. The implication is that the lateral frontal pole gets amygdala inputs via the amygdalo-fugal bundle. But the next sentence says that in monkeys the amygdala projects to the medial but not lateral frontal pole. This raises several anatomical questions.

First, how do we extrapolate from the monkey to the human lateral frontal pole? Although the monkey is our best source of cortical connectivity, the lateral pole of the monkey is considerably more primitive anatomically than humans.

This is an important consideration. As emphasized in the introduction, this study is built on the observation that “the neuronal organization and connectivity profile of the human FPI have no homologue in rodents nor other primates” (page 3). More precisely, the lateral frontal pole (FPI) is a lateral subdivision of Brodmann area 10. This area indeed has a different neural organization and connectivity fingerprint as compared to the rest of human anterior prefrontal cortex *and* macaque frontal pole (BA10). Namely, in comparison to neighboring prefrontal areas, pyramidal neurons in the FPI are relatively sparse, yet they share more dense dendritic connections (Jacobs, 2001; Ramnani and Owen, 2004). Comparative studies have shown that macaque prefrontal cortex does not contain a region with a similar connectivity profile as human lateral frontal pole (Neubert et al., 2014). Therefore, we agree with this reviewer that it would be inappropriate to extrapolate connectivity of the macaque lateral frontal pole to humans.

We have adapted the text in order to clarify the issue: page 3, lines 70-75:

More precisely, the human FPI has access to both medial and lateral cortical circuits through its extensive connections with other frontal, parietal and temporal association areas (Neubert et al., 2014). In addition, *human* FPI has direct access to information coming from the amygdala via the amygdalo-fugal bundle (Folloni et al., 2019; Bramson et al., 2020b). In contrast, macaque prefrontal cortex does not share a region homologous to human FPI, and its amygdalae project mainly to medial but not lateral prefrontal regions (Petrides and Pandya, 2007; Petrides et al., 2012).

Second, the medial frontal pole, which is part of granular PFC is often thrown together with a/dysgranular medial PFC areas, especially in relation to the default mode network. I would suggest carefully evaluating the studies cited showing amygdala connections to granular medial frontal pole to rule out the possibility that that the findings are truly about granular medial frontal pole as opposed to adjacent a/dysgranular medial PFC.

Human granular medial frontal pole (Fpm) shows a strong resemblance in terms of its connectivity profile to macaque BA 10, especially the latter's medial part (Neubert et al., 2014). This region has both efferent (Petrides and Pandya, 2007) and afferent connections with amygdala (Ghashghaei et al., 2007), although both are relatively sparse as compared to agranular medial frontal areas. Crucially, in macaques, medial BA10 receives stronger afferent as compared to efferent connections from amygdala, which separates it from a/dysgranular surrounding structures that show the opposite pattern (Ghashghaei et al., 2007). Although we cannot dissociate granular medial frontal polar cortex from adjacent a/dysgranular tissue, we think that is not a fundamental concern for this study, given its primary focus on the lateral Frontal Pole.

We have nuanced our framing of granular versus a/dysgranular -prefrontal connectivity: page 3 lines 62-70.

Contemporary models of anxiety, based on rodent studies, have shown how hippocampal-amygdala afferents to agranular medial frontal areas drive avoidance of threatening situations and fear-like behaviors (Adhikari et al., 2010, 2011), whereas recurrent medial frontal signals in the same circuit reduce threat-responses and allow approach behavior (Adhikari et al., 2010, 2015; Likhtik et al., 2014; Stujenske et al., 2014). However, extension of those rodent-based insights to human anxiety disorders has proven difficult (Likhtik and Paz, 2015; Fox and Shackman, 2019), reflected in disappointing progress in the development of new treatments for anxiety disorders (Griebel and Holmes, 2013). Those translational efforts face a major challenge in the expansion of human granular prefrontal cortex, as compared to non-human models (Pine et al., 2021; Preuss and Wise, 2022). For instance, the neuronal organization and connectivity profile of the human FPI have no homologue in rodents nor other primates (Jacobs, 2001; Ramnani and Owen, 2004; Semendeferi et al., 2011; Neubert et al., 2014).

And page 15, lines 374-378.

Although connections between agranular regions of the medial prefrontal cortex are dominated by efferent fibers that project toward the amygdala complex, BA10 shows the opposite pattern. Namely, BA10 receives stronger input from the amygdala than vice-versa²⁴. It is plausible that the same is true for lateral frontopolar cortex in humans..

Third, and related, there seems to be a contradiction about the amygdala and lateral FP. Which is correct: (1) there are no inputs from amygdala to lateral FP, or (2) the amygdalo-fugal bundle carries amygdala inputs to the lateral FP? I would suggest exploring the literature to determine the purity of amygdalo-fugal bundle. Does it have fibers from other areas besides the amygdala?

It is relevant to consider species the statements are based on.

1: There are no inputs from amygdala to lateral frontal pole *in macaques*. Mainly because macaques do not possess a homologue region to human FPI. In macaques, projections from amygdala to other lateral prefrontal areas are sparse (Petrides and Pandya, 2007).

2: The amygdalofugal bundle carries amygdala inputs to the lateral frontal pole *in humans* (Neubert et al., 2014; Folloni et al., 2019; Bramson et al., 2020b).

We are not aware of any other regions that project to the frontal pole via the amygdalofugal pathway. We have carefully reviewed the paper for clarity on the species on which anatomical statements are based. And added 'human' at several instances.

If all of the above is clarified, I would want to know more about which regions of the amygdala connect to lateral and medial FP. In general, the basolateral areas of the amygdala connect with neocortex and the central and medial regions connect with mesocortical, agranular medial cortex. Lateral and medial FP, being part of granular neocortex, might be expected to be connected with the basolateral amygdala. Knowing these details will be very important in understanding if and how amygdala connections with polar PFC contribute to anxiety.

Amygdala fibers projecting to the prefrontal cortex mainly stem from the basal or lateral nucleus (Ghashghaei et al., 2007; Aggleton et al., 2015), as these are the main output nuclei projecting to cortex. More specifically, the fibers projecting to BA10 in macaque originate mostly from the magnocellular division of the basal nucleus (Aggleton et al., 2015). At present, it is unknown whether this holds for human lateral Frontal Pole, but it is parsimonious to assume that this connectivity pattern of amygdala nuclei is preserved.

Interestingly, in macaques, amygdala connections with BA10 are dominated by bottom-up amygdala→BA10 projections rather than BA10→amygdala, a pattern that is opposite to that observed in agranular medial prefrontal cortices (Ghashghaei et al., 2007; Petrides and Pandya, 2007) but in line with the notion of the amygdalofugal path being the main output pathway of the amygdala complex.

As for the amygdalofugal pathway as a whole, it contains fibers stemming from basolateral but also - to a lesser extent - from the central nuclei of the amygdala, although the fibers stemming from the central nucleus probably do not extend all the way to prefrontal cortex but instead project to the Bed nucleus of the Stria Terminalis (BST) (Fox and Shackman, 2019).

We have added this information to the discussion, page 15, lines 368-377:

The amygdalofugal bundle contains fibers stemming from both basal and lateral amygdala nuclei as well as the central nucleus. Although it is unclear where the amygdalofugal projections to FPI originate specifically, it is likely that these projections stem from the basal or lateral nucleus, given that most of the central nucleus projections terminate in the Bed Nucleus of the Stria Terminalis (Fox and Shackman, 2019), and the basal nucleus is considered the main output node of the amygdala complex (Aggleton et al., 2015). It is possible that the projections to FPI are an extension of magnocellular basal nucleus projections that, in macaque, extend to BA10 (Fpm in humans) (Aggleton et al., 2015). Although connections between agranular regions of the medial prefrontal cortex are dominated by efferent fibers that project toward the amygdala complex, BA10 shows the opposite pattern. Namely, BA10 receives stronger input from the amygdala than vice-versa (Petrides and Pandya, 2007).

My other concern is about what anxiety itself is. For the anxious person, much of what it means to be anxious centers on how the patient feels subjectively. The only hint of concern about the feeling of anxiety in the paper is the citation of Glassman 2014 study on public speaking. I think the paper could be strengthened by including relevant citations about the FP and conscious experience in general and

conscious fear and anxiety. Some suggestions follow.

Bang D, Ershadmanesh S, Nili H, Fleming SM. Private-public mappings in human prefrontal cortex. *Elife*. 2020 Jul 23;9:e56477. doi: 10.7554/eLife.56477. PMID: 32701449; PMCID: PMC7377905.

Brown R, Lau H, LeDoux JE. Understanding the Higher-Order Approach to Consciousness. *Trends Cogn Sci*. 2019 Sep;23(9):754-768. doi: 10.1016/j.tics.2019.06.009. Epub 2019 Jul 30. PMID: 31375408.

Cushing CA, Dawes AJ, Hofmann SG, Lau H, LeDoux JE, Taschereau-Dumouchel V. A generative adversarial model of intrusive imagery in the human brain. *PNAS Nexus*. 2023 Jan 23;2(1):pgac265. doi: 10.1093/pnasnexus/pgac265. PMID: 36733294; PMCID: PMC9887942.

Lau H, Rosenthal D. Empirical support for higher-order theories of conscious awareness. *Trends Cogn Sci*. 2011 Aug;15(8):365-73. doi: 10.1016/j.tics.2011.05.009. Epub 2011 Jul 6. PMID: 21737339.

LeDoux JE. What emotions might be like in other animals. *Curr Biol*. 2021 Jul 12;31(13):R824-R829. doi: 10.1016/j.cub.2021.05.005. PMID: 34256909.

Taschereau-Dumouchel V, Michel M, Lau H, Hofmann SG, LeDoux JE. Putting the "mental" back in "mental disorders": a perspective from research on fear and anxiety. *Mol Psychiatry*. 2022 Mar;27(3):1322-1330. doi: 10.1038/s41380-021-01395-5. Epub 2022 Jan 26. PMID: 35079126; PMCID: PMC9095479.

We agree that the subjective experience is a crucial component in disorders such as anxiety. In our case, we have included high-anxious individuals based on their self-reported anxiety, assessed via the LSAS questionnaire. We then used the STAI questionnaire to assess experienced symptoms, and those symptom scores correlate both with FPI involvement in the AA task and strength of the amygdalofugal projections to FPI. We have extended the discussion on anxiety symptoms and incorporated several of the suggested papers into the manuscript.

Discussion page 13-14, lines 330-332.

Putatively, the overexcitable FPI we observe in high-anxious, combined with stronger amygdala afferences when controlling emotional actions might make it difficult for anxious individuals to maintain their private convictions when conforming to social norms, a role attributed to FPI⁴².

Discussion page 15, lines 378-386.

Our results support recent suggestions that FPI arbitrates between imagined and veridical threat on the basis of magnocellular inputs(Cushing et al., 2023). Increased amygdala projections might make it difficult for anxious individuals to correctly attribute the assessed dangers projected to FPI to imagined or veridical threats. FPI's potential role as an arbitrator in threat imagery has mostly been described in terms of potential involvement of intrusive memories in PTSD (Cushing et al., 2023) and more generally fits recent views on the role of the FPI in emotional experience, such as anxiety, and its regulation (LeDoux, 2020). Interestingly, increased FPI activation during emotion control in our task can protect against the development of PTSD symptoms after trauma(Kaldewaij et al., 2021), and exposure therapy

has been shown to restore frontopolar function in those PTSD patients that benefit from treatment(Fonzo et al., 2017).

Reviewer #2 (Remarks to the Author):

I am reviewing the paper “Anxious individuals shift emotion control from lateral frontal pole to dorsal prefrontal cortex; functional, structural and neurochemical evidence” by Bramson et al. In this study, the authors examine lateral frontopolar (FPI) function in control of emotional action using a well validated task and a multi-modal imaging approach, including functional neuroimaging, structural connectivity (DWI), and spectroscopy metrics that reflect the baseline excitability (GABA/Glu ratio) of FPI and PFC regions of interest. The authors contrast FPI & (DL)PFC activation profiles between high-anxious vs. non-anxious individuals, and find that (1) Anxious participants show less “emotion incongruency” activation in FPI compared to non anxious; (2) Anxious participants show instead increased congruency-sensitive activation in dorsolateral PFC (DLPFC) compared to non-anxious individuals; (3) The magnitude of functional engagement of FPI to the incongruency contrast is inversely correlated with anxiety scores, further corroborating finding (#1). Moreover, the authors attempt to relate these distinct profiles of FPI engagement in their task to other trait-like variation in the structure and excitability of FPI. Here, they find that (4) FPI is more excitable in anxious vs. non-anxious individuals; and the magnitude of FPI excitability is differentially associated with the (behavioral) congruency effect in their task as a function of anxiety (such that non-anxious individuals have better emotional control with more excitable FPI; but vice versa for anxious folks). There are additional findings here that were not readily intuitive to me, which included that (5) the behavioral congruency effect was related to GABA/Glu in SMC in the non-anxious group, and that (6) the neural congruency effect in SMC correlated with FPI excitability only in the non-anxious group. When examining FPI structure, the authors found that (7) amygdalofugal pathway/projections to FPI were stronger in high-anxious individuals and that this group difference was spatially specific to FPI and not amygdalofugal projections to medial PFC (BA24/25); and (8) that the association between amygdalofugal anatomy and emotional control changed between the groups, such that in the non-anxious group only, higher structural connectivity was associated with greater behavioral congruency effects (worse control). They also note that (9) high anxiety participants show associations between amygdalofugal strength and neural congruency effect in mPFC and ACC. Finally, the authors find that amygdalofugal-FPI strength was associated with greater DLPFC, further corroborating finding (2).

On one hand, the quest of determining regional specificity in the PFC for the cognitive control of emotion is timely and in critical need of new data and careful analysis aimed at that goal. The authors have amassed an impressive amount of data to that end and have prioritized intra-subject reliability (collecting a large number of trials per subject; $n > 500$), which is commendable and rarely done in affective neuroscience. Moreover, they use a task that is well validated by the past decade of their work, a body of work that had raised important questions regarding FPI “helping” or “hindering” regulation, which the present manuscript helps to answer. On the other hand, there were aspects of the current manuscript that I found challenging to parse—conceptually, and I also had a few methodological concerns, which if addressed, could strengthen an already impressive body of work.

Major concerns

1. For finding in Figure 2C, I appreciate using Bayesian statistics, but another strong and easily interpretable test is whether there is a significant interaction between Group and the Congruency effect. Same question for Figure 2D: in other words, is FPI (incongruent) significantly more engaged for Non-anxious vs. Anxious, and DLPFC significantly more engaged for Anx vs. Non-Anx?

We agree with the reviewer that a simple interaction analysis is important for interpretability of the findings displayed in Figure 2 and we indeed report it in the manuscript: Our whole brain and frontal-cortex analyses contrasted congruency effects between high-anxious and non-anxious participants. On page 6 we report that there is a stronger congruency effect in high-anxious in dPFC (figure 2D) but no difference in FPI. However, because we did not observe a congruency effect in FPI within the high-anxious group, an effect consistently observed in multiple earlier studies in non-anxious samples (Volman et al., 2011, 2013; Tyborowska et al., 2016; Bramson et al., 2018, 2020b; Kaldewaij et al., 2019a, 2021; Koch et al., 2019), we employed Bayesian statistics to assess the evidence for no effect. The Bayesian tests can be interpreted as a clarification of a null result observed in the Group by Congruency test the reviewer suggests for FPI.

We have changed our phrasing in the results section to clarify the logic of analyses: page 6, lines 139-146.

There was no interaction between group (high vs non-anxious) and congruency in FPI when correcting for voxels across the whole frontal cortex. However, separating this analysis for non-anxious and high-anxious participants showed significant FPI activation in non-anxious but no statistically reliable congruency effect in the FPI for high-anxious participants, suggesting that they might rely less on FPI for control, figure 2C. To assess this possibility we used Bayesian t-test to clarify potential absence of FPI recruitment high-anxious individuals in the specific FPI territory recruited in healthy controls (figure 2C black circles) confirmed this observation, providing moderate evidence for the absence of this effect in the high-anxiety group, $BF_{01} = 4.2$.

2. An additional (potentially more direct) test of the authors' interesting conjecture regarding DLPFC vs. FPI engagement in the High Anx group is that the magnitude of Inc>Cong in DLPFC should be inversely correlated with the magnitude of Inc>Cong FPI, either in High Anx alone, or potentially across both groups. In other words, if DLPFC engagement is reflecting some sort of compensatory effect, then it should be inversely related to how much FPI is 'up to the task'. Is that the case? And/or is the delta DLPFC-FPI across subjects related to anxiety and/or the emotional congruency effect?

Across the two groups we observed a negative correlation between dPFC and FPI congruency effects, $\rho(91) = -.22$, $r = .038$, indeed confirming the reviewers suggestion and our speculation that the two are inversely related. Considering this relationship within each group separately suggests that this correlation is mainly driven by the anxious group, as those show a similar correlation between FPI and dPFC congruency, although not statistically reliable; $\rho(50) = -.2$, $p = .15$. The non-anxious group does not show relationship between FPI and dPFC neural congruency; $\rho(39) = -.06$, $p = .7$.

We have added those findings to the results section: page 6, lines 156-160.

Interestingly, across the groups those participants that recruited FPI the least relied most on dPFC, $\rho(91)=-.22$, $r = .038$, again suggesting that dPFC compensates for reduced recruitment of FPI. Numerically, this relationship is driven by the high-anxious individuals $\rho(50)=-.2$, $p = .15$, as compared to the non-anxious; $\rho(39) = -.06$, $p = .7$, although tested separately, both were not statistically reliable.

Further, the dPFC neural congruency effect correlates differently with behavioral congruency between groups (Congruency*Group*dPFC BOLD effect interaction). In the anxious group, those participants with stronger dPFC activation in incongruent versus congruent conditions perform better; $\rho(50)= -.28$, $p=.04$, this is not the case for the non-anxious group; $\rho(39)= 0$, $p=.98$.

We have added this information to the results section, page 6-7 lines 160-163 .

Behavioral congruency was also differentially related to neural congruency in dPFC between groups $b = .11$ [$.02 .19$]. In the high-anxious neural congruency effects in dPFC correlated negatively with behavioral congruency; $\rho(50)= -.28$, $p=.04$, whereas this is not the case for the non-anxious participants, $\rho(39)= 0$, $p=.98$.

3. The analysis that included Left SMC (Figures 3C & 3D) (also, why Left and not Right?) did not seem theoretically motivated until I got deep into the Methods section. If the authors consider that set of findings of equal import to the manuscript (vs. being in Supplementary) compared to FPI/DLPFC findings, then additional setting up of the theoretical framework—including prior findings—and stating the hypotheses in the Introduction would be warranted. As is, I found those findings quite difficult to follow and to integrate with FPI and DLPFC results.

This is an important point also brought forward by reviewer #3. The reason we take left sensorimotor cortex, rather than right SMC, is because our participants make approach and avoidance actions by moving a joystick towards or away from themselves with their right hand.

We agree that the manuscript would benefit from better introducing the role of FPI-SMC connectivity during emotional action control: In previous studies we have shown that when people select affect-incongruent actions, increases in high-frequency neural activity in left SMC are phase-synchronized with low-frequency oscillations in right FPI (Bramson et al., 2018). Improving effective connectivity between FPI and SMC using dual-site tACS (in-phase vs anti-phase) can improve emotional-action control (Bramson et al., 2020a). Those observations were the starting point for the current studies. Therefore, we included left SMC as a region of interest for GABA/Glx voxel placement and consequent analyses. The findings reported in figures 3C&D specifically concern the relationship between FPI and SMC, not SMC alone. As we believe the FPI is involved in implementing emotional-action control by influencing left SMC in the current task, and FPI involvement is putatively reduced in the high-anxious individuals, this would likely result in a decoupling of FPI-SMC. The evidence supports this possibility, as visualized in figures 3C and 3D. We agree that this background could be better grounded in the introduction and results sections, which we have adapted accordingly.

Introduction: page 3, lines 75-77:

Accordingly, recent work has suggested that FPI is involved in selecting emotional actions by influencing neural activity in sensorimotor cortex (SMC) when different alternative options are available (Bramson et al., 2018; Koch et al., 2018).

Results: page 8, lines 194-198

We acquired MRS scans from right FPI (figure 3A), left SMC and left occipital lobe (Supplementary figure 2). The first two locations are known to support the implementation of emotion control (Bramson et al., 2018), and the latter location provides a control region. Left SMC was selected because it has been shown to be under FPI control when selecting affect-incongruent movements with the right hand (Bramson et al., 2018, 2020a).

Page 9, lines 215-217.

This interaction complements earlier findings using the same experimental paradigm, showing that FPI interacts with left SMC to implement control over affect-incongruent actions that are performed with the right hand (Bramson et al., 2018, 2020a).

4. I found the reverse association between Gaba/Glu and the behavioral congruency effect intriguing (Fig 3B). It suggests there's a point at which FPI function/engagement is 'counter-productive' for Anx folks. I would have liked to see those findings more directly connected to DLPFC function (Fig 2B)—are these two separate stories—i.e. are these effects impacting different individuals? Or are the High Anx individuals with more excitable FPI (less helpful) also the ones recruiting DLPFC more during the task (Inc > Con)? Can you enter those IVs (GABA/Glu, DLPFC engagement, FPI engagement, amygdalofugal pathway-FPI) in the same simultaneous regression model predicting Anx symptoms or the behavioral congruency effect to ascertain whether these neural metrics are explaining overlapping variance, or possibly exerting 'suppression' (in a regression framework) effects?

(On suppression, see: <https://www.ncbi.nlm.nih.gov/pmc/articles/PMC3780394/>)

We have taken several steps to answer this question.

First, we have performed a Bayesian mixed effects model explaining performance based on a six-way interaction between *Congruency (congruent/incongruent) * Group (non/high-anxious) * FPI BOLD * dPFC BOLD * FPI GABA/Glx * amygdalofugal-FPI connectivity*. This model results in a significant *Congruency * Group * FPI GABA/Glx* interaction. However, several interactions we observed earlier are no longer statistically reliable. Given that *Congruency * Group * amygdalofugal connectivity* (figure 4B), and *Congruency * Group * dPFC BOLD* were significant when considered in isolation, we infer that *dPFC congruency* and *amygdalofugal-FPI connectivity* explain shared variance in the congruency effects on behavior, as the reviewer suggests. Accordingly, amygdalofugal-FPI projections correlate to dPFC engagement (figure 4D).

However, while *amygdalofugal-FPI connectivity* and *dPFC BOLD congruency* explain shared variance, this shared variance is at least partly separate from the variance explained by *FPI GABA/Glx*. This is intuitive given that GABA/Glx ratio in FPI explains more variance in behavior in non-anxious (figure 3B), whereas high-anxious recruit dPFC rather than FPI. The extent of this compensatory recruitment depends on the strength of the amygdalofugal projection strength (Figure 4D).

Given that the study was not set up to test six-way interactions (and therefore lacks statistical power), and given the relatively indirect nature of these observations, we have added these findings to the supplementary information section.

Shared variance in behavior explained by separate regressors

To assess which regressors might explain shared variance in the behavioral congruency we performed a Bayesian mixed effects model explaining performance based on a six-way interaction between *Congruency (congruent/incongruent) * Group (non/high-anxious)*FPI engagement (BOLD effect congruentVSIncongruent) * dPFC engagement * FPI GABA/Glx * amygdalofugal-FPI connectivity*. This model results in a significant *Congruency*Group*FPI GABA/Glx* interaction; $b = .2$, $CI [.04 .37]$. However, several interactions we observed earlier are no longer statistically reliable. Given that *Congruency*Group*amygdalofugal connectivity* (figure 4B), and *Congruency*Group*dPFC BOLD* were significant when considered in isolation, we infer that *dPFC congruency* and *amygdalofugal-FPI connectivity* explain shared variance in the congruency effects on behavior. Accordingly, amygdalofugal-FPI projections correlate to dPFC engagement (figure 4D).

However, while *amygdalofugal-FPI connectivity* and *dPFC BOLD congruency* explain shared variance, this shared variance is at least partly separate from the variance explained by *FPI GABA/Glx*. This is intuitive given that GABA/Glx ratio in FPI explains more variance in behavior in non-anxious (figure 3B), whereas high-anxious recruit dPFC rather than FPI. The extent of this compensatory recruitment depends on the strength of the amygdalofugal projection strength (Figure 4D).

5. The gender imbalance across the samples raises questions about how to best address this confound. It was not clear to me exactly how Figure 4B or Suppl. Figure 3 addressed it (though they're mentioned in the Discussion). Is there an additional control sample of non-anxious individuals that includes females that the authors could use for at least a subset of these analysis? (e.g. structural); or vice versa (i.e. a high anxiety group with male subjects only). The uncinate fasciculus (for instance, another important amygdala pathway) is known to show gender differences:

<https://ajp.psychiatryonline.org/doi/full/10.1176/appi.ajp.2018.18040425>

<https://www.frontiersin.org/articles/10.3389/fnsys.2016.00093/full>

If no additional (balanced) samples available, can the authors control for gender in their model to see if results hold?

This is an important caveat (see also comment #4 by reviewer #3) that was not addressed in the figures. As only 13 males are included in the high-anxious sample, it is not possible to reliably consider gender by adding it as a factor in our statistical model, and currently we do not have access to a suitable control sample (e.g. non-anxious females or high-anxious males) that also includes DWI or MRS data.

We have amended the discussion to more clearly point to potential caveats in structural connectivity due to gender differences pointed out by the reviewer. However, other work has shown that microstructural properties of amygdalofugal pathway do not seem to differ between sexes. Whether this is the case for amygdalofugal projections to lateral frontopolar cortex remains to be determined.

Page 15-16, lines 387-405.

It could be argued that the current findings are biased by the skewed distribution of males and females across the high-anxious and non-anxious groups. For the neural and behavior congruency effects this is unlikely, as both male-only^{3,30} and female only⁶³ studies using the AA task have shown FPI recruitment, and large scale mixed samples did not find differences between males and female

participants in FPI engagement^{9,64,65}. We also consider it unlikely that males and females differ in specific characteristics of FPI linked to emotional-action control, such as the structural connections from amygdala to FPI and FPI neurochemical profile, in the context of group-matched amygdalofugal projections to medial prefrontal cortices and excitability in SMC and V1 (Figure 4C & supplementary figure 2A;B). Although gender differences in the development of GABA concentration across the lifespan have been reported⁶⁶, large scale studies comparing male and female participants did not show gender differences in the relationship between GABA and Glx in posterior⁶⁷, or prefrontal cortex⁶⁸. Further, although absolute levels of GABA might be different between males and females, the relationship between GABA and glutamate does not seem to vary across gender⁶⁹. Gender differences have been shown in the relationship between anxiety and white-matter connectivity between amygdala and prefrontal cortices^{70,71}. However, these were based on whole-bundle average estimates of structural integrity in the Uncinate Fasciculus, rather than differences in relative strength of projections. Microstructural assessment of amygdalofugal white-matter properties do not differ between males and females⁷². Future studies could more stringently test the potential influence of gender differences to amygdalofugal connectivity and FPI neural excitability.

6. Supplementary Figure 3 is quite novel and important; I was surprised to see it in Supplementary. In particular, I would be interested in knowing whether there is an interaction between region (Fpl vs. BA25 and 24) and Group (as for BA24/25 the association with Anx seems numerically reversed). It is also interesting that the amygdalofugal_FPI pathway correlates positively with anxiety, which warrants additional discussion.

This is an interesting suggestion, in particular given the knowledge on the difference in dominance of efferent versus afferent projections between area 24/25 and FPM (BA10) in macaques. Whereas area 24 and 25 have relatively more efferent connections towards the amygdala complex, FPM is a net receiver of projections from amygdala (Petrides and Pandya, 2007). However, we did not observe a region by group interaction in this dataset. The correlation between anxiety and amygdalofugal tract strength to FPI was placed in the supplements because we reasoned it was mainly a support for the group*amygdalofugal tract differences and its interaction with behavior shown in figure 4A;B. We have moved these results to figure 4 in the main manuscript.

In addition we discuss the relation between anxiety and the strength of the amygdalofugal-FPI path now in the discussion. Page 13-14, 330-332.

Putatively, the overexcitable FPI we observe in high-anxious, combined with stronger amygdala afferences when controlling emotional actions might make it difficult for anxious individuals to maintain their private convictions when conforming to social norms, a role attributed to FPI⁴².

7. Figure 4D is very helpful for connecting and interpreting the structural-functional findings across the paper; but here I found Figure 4C difficult to integrate with the rest. Is there spatial overlap between 4C and 4D? How does 4C (medial PFC, also dorsal PFC, interaction with Group and amygdalofugal pathway) help with the interpretation of 4D (greater DLPFC engagement across groups for individuals with greater amygdalofugal pathway to FPI)?

We agree that Figure 4D is more helpful than Figure 4C. Because the group differences in the interaction between amygdalofugal projections to FPI and neural congruency is further specified in the visualization

in supplementary figure 3, we deem 4C superfluous and removed it. This creates space for showing data on the amygdalofugal projections to different medial prefrontal areas (point #6).

Minor points

1. Intro Line 77: 'deviant recruitment'- awkward phrasing

We thank the reviewer for pointing to this phrase. The sentence now reads:

Based on these findings, we reasoned that aberrant FPI recruitment might account for the difficulties experienced by individuals with anxiety in situations where they need to control emotional action tendencies.

2. Intro Line 95/Discussion Line 282: I am not sure the present data permit strong inferences regarding FPI afferents vs efferents; can the authors elaborate on this?

The reviewer is correct in their assessment, we cannot infer direction from the current data. Our suggestions were based on monkey granular prefrontal cortex mainly receiving amygdala efferences, although it also contains afferent fibers (Petrides and Pandya, 2007). Combined with our observation that increased amygdalofugal projections interfere with emotional-action control, we hypothesized stronger influence of affective information in action selection (Bramson et al., 2020b). However, this is conjecture.

We have toned down those claims in the introduction and discussion, page 4, lines 96-98:

Furthermore, stronger amygdala connections, in the context of reduced FPI neuronal responsivity, significantly accounted for the anxiety-related shift towards those alternative control circuits in the frontal lobe.

Page 13, lines 308-313

There are three main findings. First, anxious individuals use dPFC, rather than FPI as their non-anxious peers, to implement control over emotional action tendencies. Second, FPI in anxious individuals might receive stronger input from the amygdala via more extensive amygdalofugal pathway connections, and the magnitude of that structural connection predicts the degree of FPI-dPFC shift during the implementation of emotional control.

3. Is there spatial overlap between the results shown in 2E and in 2C? If so, that would be worth indicating.

We have added a figure to the supplements showing the overlap between the neural congruency effects in the non-anxious (that show FPI activation) and the correlation between congruency effects and trait-anxiety. See supplementary figure 1B.

4. I believe there is an error when referring to Figure 4; there is no 4E and I think the lettering is off by one throughout.

We have corrected this mistake throughout the manuscript.

5. I do wonder whether the authors examined differential amygdala activation profiles across these Anx vs. Non Anx groups and whether those relate to FPI structural, neurochemical, or functional profiles?

Potential differences in amygdala activity between groups, and their relationship with our variables of interest could have been seen in whole-brain analyses. We did not see any indication of group-differences in any of those tests.

6. This is at surface minor but it is important for interpretation. At times the ‘congruence’ effects seemed to refer to FPI’s increased engagement to incongruent vs. congruent, but later it was referred to primarily as the facilitation of action by emotion. Keeping the labeling, terminology (and contrast direction) consistent throughout the manuscript would facilitate parsing through (and connecting) the present results.

We thank the reviewer for pointing this out and now consistently refer to ‘neural congruency’ or ‘behavioral congruency effects’ throughout the manuscript.

Reviewer #3 (Remarks to the Author):

Bramson et al. investigate the role of the brain circuit involving the lateral frontolpolar cortex (FPI), posterior parietal cortex, sensorimotor cortex and amygdala in emotion-action control in subjects with anxiety. They used magnetic resonance spectroscopy, diffusion weighted imaging and functional MRI to target FPI GABA/Glx levels, functional activation of FPI during a mild emotional challenge requiring emotional action control in high-anxious (n=52, 14 male) and non-anxious (n=41, all male) participants. Their main findings are the following: relative to non-anxious subjects, high-anxious participants use more dPFC in emotion-action control, have lower FPI GABA/Glx, and show stronger connectivity in amygdalofugal projection to FPI. They conclude that there is a functional-anatomical shift in emotion-action control in anxious subjects, and this shift is associated with the change in structural connectivity of amygdalofugal projections to FPI and overexcitability in FPI.

Overall, I find that it is very nice to use a multimodal approach addressing the question from functional, structural and neurochemical aspects, which may provide mechanistic insights. I have a few major comments related to the study methodologies and design, which may affect the results, interpretation and conclusion.

Major comments:

1. A “statistical analysis” section is missing in the method part. For example, a lot of 3 or 4 way interaction analysis were done also with some post-hoc analysis. But it is not clear whether the interaction is significant or not. If not, then the post-hoc analysis may be not valid. Was multiple comparison correction applied? Which kind of correction was used? A summary table with statistical results will be useful to have an overview of the results.

We agree that the paper would benefit from a statistical analysis section in the methods. The significant 3 and 4 way interactions are described in the text. To clarify those further, we describe them in the

added statistical analysis paragraph in the methods section and a supplementary table that shows the models tested and the most important lower-order interactions. Most analyses were covered by separate hypotheses. Multiple-comparisons Bonferroni correction was used to correct across the three regions indexed with MRS when interpreting GABA/Glx ratio correlations with behavior (Supplementary figure 2B). Whole-brain results were corrected for multiple comparisons for all voxels in the brain or in the frontal lobe using cluster-correction using a cluster-forming threshold of $z > 2.3$.

Page 21, lines 608-631

Analyses – statistics

Statistical models testing behavioral congruency effects across and between groups, and derived models adding covariates were run in a step-wise fashion. We first compared correct responses between congruent and incongruent conditions between the different groups: Group (non-anxious vs high-anxious) * congruency (congruent vs incongruent). We then extended this model in two iterations by adding estimates of excitability in a 4-way interaction: Group*congruency* FPI GABA/Glx * SMC GABA/Glx. Amygdalofugal tract strength was added in a separate three-way interaction: Group*Congruency*amygdalofugal tract strength. Significant interactions were assessed by interpreting lower-level interactions resulting from these same models, or post-hoc Spearman correlations. Full models and results for the most important interactions are presented in supplementary table 4. Correlations between neural excitability and behavioral congruency for the different groups were Bonferroni corrected for the three regions of interest. Analyses on functional MRI effects were cluster corrected using a cluster-threshold forming threshold of $z > 2.3$ controlling either for all voxels in the brain (whole brain analyses) or all voxels in the frontal lobe.

2. For neurochemical data measured by MRS, authors used GABA/Glx ratio to evaluate the excitability in FPI. The increased excitability is concluded from reduced GABA/Glx. In fact, reduced GABA/Glx can be led by increases in Glx or decreases in GABA, or decreases in both but with more extent in GABA. Therefore, authors should quantify GABA and Glx separately, which will facilitate the interpretation of the results. Did authors acquire unsuppressed water spectra? You can quantify GABA and Glx separately, instead of using GABA/Glx. In this case, one can know if the change observed is driven by e.g. reductions in GABA or just increases in Glx.

Although our hypothesis was based on the GABA/Glx ratio (see also refs) we agree it is interesting to test for individual GABA and Glx effects. We only acquired MRS spectra with water suppression. It is therefore not possible to scale GABA and Glx estimates to water, giving a 'clean' estimate of both chemicals. However, we have added analyses relating both GABA and Glx estimates to Creatine (Cr), also present in our spectra. There is no statistical difference between non-anxious and high-anxious groups in either GABA/Cr and Glx/Cr ratio, suggesting that the ratio between inhibition and excitation is the important driver of our effects, not GABA or Glx in isolation. The 4-way interaction explaining behavioral performance (Group*Congruency*FPI excitability * SMC excitability), and the three-way interaction between Group*Congruency*FPI excitability also do not hold when testing GABA/Cr and Glx/Cr separately. Furthermore, neither GABA/Creatine nor Glx/Creatine correlate with behavioral congruency

We have added this to the results section, page 9, lines 223-230.

To assess whether the results presented above can be attributed specifically to GABA or Glx alone, we repeated the main analyses by considering GABA and Glx independently, as a proportion of Creatine concentration. There was no difference in either FPI GABA/Cr ratio: $t(89) = 1.14, p = .25$, or FPI Glx/Cr ratio: $t(89) = .29, p = .77$ between groups. There were also no correlations between behavioral congruency effects and FPI GABA/Cr or Glx/Cr ratios; all $\rho < .2, p > .16$. Combined, these results suggest that the ratio between GABA and Glx is important for FPI-based emotional action control, and that it is specifically the ratio between inhibition and excitation in FPI that is different in high-anxious as compared to non-anxious individuals.

3. Most of the mechanistic interpretation in the discussion is established on increased excitability in the FPI of anxious subjects. As commented above, these discussion part should be revised after looking into individual GABA and Glx results.

Although the effects are indeed not the consequence of isolated effects of GABA or Glx but to the GABA/Glx ratio, we added discussion on the increased excitability interpretation. Page 16, lines 414-417.

The relationship between GABA/Glx ratio and behavioral congruency could not be attributed to effects of GABA (vs creatine) or Glx (vs creatine alone, suggesting emotional action control depends on relative inhibition/excitation in lateral frontal pole, rather than inhibitory or excitatory tone as such.

4. Another big issue is that the gender is largely biased between high-anxious (only 14 males) and non-anxious (all males) groups. The authors mentioned that gender has minor effect on the functional FPI engagement. They also claim that the structural connections and FPI neurochemical profile are unlikely affected by gender, which is lack of supporting evidences. At least, the gender effect on brain metabolites has been reported. If gender does have an effect on GABA and Glx, then authors should be careful that the observed difference may be dominated by the gender difference.

We agree that we lack appropriate evidence for concluding that our results are not biased by gender and have toned down those claims, see also our response to reviewer #2. Further, we have added a discussion on potential gender differences in brain metabolites to the manuscript. Page 15-16, lines 387-405.

It could be argued that the current findings are biased by the skewed distribution of males and females across the high-anxious and non-anxious groups. For the neural and behavior congruency effects this is unlikely, as both male-only^{3,30} and female only⁶³ studies using the AA task have shown FPI recruitment, and large scale mixed samples did not find differences between males and female participants in FPI engagement^{9,64,65}. We also consider it unlikely that males and females differ in specific characteristics of FPI linked to emotional-action control, such as the structural connections from amygdala to FPI and FPI neurochemical profile, in the context of group-matched amygdalofugal projections to medial prefrontal cortices and excitability in SMC and V1 (Figure 4C & supplementary figure 2A;B). Although gender differences in the development of GABA concentration across the lifespan have been reported⁶⁶, large scale studies comparing male and female participants did not show gender differences in the relationship between GABA and Glx in posterior⁶⁷, or prefrontal cortex⁶⁸. Further, although absolute levels of GABA might be different between males and females, the relationship between GABA and glutamate does not seem to vary across gender⁶⁹. Gender differences have been shown in the

relationship between anxiety and white-matter connectivity between amygdala and prefrontal cortices^{70,71}. However, these were based on whole-bundle average estimates of structural integrity in the Uncinate Fasciculus, rather than differences in relative strength of projections. Microstructural assessment of amygdalofugal white-matter properties do not differ between males and females⁷². Future studies could more stringently test the potential influence of gender differences to amygdalofugal connectivity and FPI neural excitability.

Minor comments:

1. Why left SMC, and right FPI and occipital lobe were investigated? Why not stay with the same side?

Please see our response to reviewer #2, point #3:

The reason we take left sensorimotor cortex, rather than right SMC, is because our participants make approach and avoidance actions by moving a joystick towards or away from themselves with their right hand.

We agree that the manuscript would benefit from better introducing the role of FPI-SMC connections during emotional action control: In previous studies we have shown that when people select affect-incongruent actions, increases in high-frequency neural activity in left SMC are phase-synchronized with low-frequency oscillations in right FPI (Bramson et al., 2018). Improving this effective connectivity using dual-site tACS (in-phase vs anti-phase) can improve emotional-action control (Bramson et al., 2020a). As this was the starting point for the current studies, we included left SMC as a region of interest for GABA/Glx voxel placement and consequent analyses. The importance of the findings reported in figures 3C&D specifically concern the relationship between FPI and SMC, and not SMC alone. As we believe the FPI is involved in implementing emotional-action control by influencing left SMC in the current task, and FPI involvement is putatively reduced in the high-anxious individuals, this would likely result in a decoupling of FPI-SMC. This is indeed the case, which we have attempted to visualize in figures 3C and 3D, albeit reflected in this indirect estimate of interactions between FPI and SMC baseline excitability and behavioral congruency. We agree that this background could be better grounded in the introduction and results sections, which we have adapted accordingly.

Introduction page 3, lines 75-77:

Accordingly, recent work has suggested that FPI is involved in selecting emotional actions by influencing neural activity in sensorimotor cortex (SMC) when different alternative options are available (Bramson et al., 2018; Koch et al., 2018).

Results page 8, lines 194-198:

We acquired MRS scans from right FPI (figure 3A), left SMC and left occipital lobe (Supplementary figure 2). The first two locations are known to support the implementation of emotion control (Bramson et al., 2018), and the latter location provides a control region. Left SMC was selected because it has been shown to be under FPI control when selecting affect-incongruent movements with the right hand (Bramson et al., 2018, 2020a).

Page 9, lines 215-217

This interaction complements earlier findings using the same experimental paradigm, showing that FPI interacts with left SMC to implement control over affect-incongruent actions that are performed with the right hand (Bramson et al., 2018, 2020a).

2. Please specify the snr in MRS data, is it the snr of edited spectra (referring to GABA peak? Or Glx peak?) or non-edited spectra?

We apologize for the unclarity. Snr refers to the GABA peak in the edited spectra. We have added this information to the table description:

This table shows GABA and Glx data quality measures for all three voxels; Cramer-Rao lower bound (Sd); Full width half maximum of the spectrum (FWHM); and estimated signal to noise (Snr) for the GABA peak extracted from the edited-spectrum, all provided by LCmodel. We excluded 3 FPI spectra based on a combination of Cramer-Rao lower bound, low signal-to-noise and visual inspection(Kreis, 2016).

3. What is the specific parameter measured from DWI for “connectivity”?

This is indeed an unhelpful label. We refer to the percentage of times probabilistic tractography ended up in the anatomical region FPI. We have changed this label in the figures to *amygdalofugal-FPI connectivity*.

Reviewers' Comments:

Reviewer #1:

Remarks to the Author:

I am satisfied with most of the changes. However, I suggest that the authors include more citations to the subjective feelings of anxiety, such as some of the ones I suggested or others.

Reviewer #2:

Remarks to the Author:

I am reviewing the first revision of the paper "Anxious individuals shift emotion control from lateral frontal pole to dorsal prefrontal cortex; functional, structural and neurochemical evidence" by Bramson et al. The authors have addressed most of my concerns, but a few points of concern remain, which are detailed below:

- (1) Lines 139-146: Using a test of the null hypothesis in the absence of a significant Group X Congruency interaction (Figure 2C) seems unusual. That paragraph would flow better if the authors first detailed the nature of the FPI effect in the non-anxious group, and then tested the interaction (and the Bayesian null).
- (2) The lack of interaction (differences in slopes) for results now reported in lines 158-159 & 162-163 is concerning and reduces their interpretability. Apart from "(...)those 156 participants that recruited FPI the least, relied most on dPFC, $p(91)=-.22$, $r = .038$, again suggesting that dPFC compensates for reduced recruitment of FPI", I would suggest removing them to avoid unwarranted interpretation of group differences in the absence of slope differences. (Reporting that they are n.s. would suffice). Clarify what $b = .11$ [.02 .19] reflects.
- (3) Line 330: Missing word: "(...) the overexcitable FPI we observe in high-anxious (?...)"
- (4) Nomenclature re: dorsal prefrontal cortex-> given this is referring to 9-46, wouldn't it be more anatomically specific to refer to this region as dorsolateral PFC (DLPFC) instead of just 'dorsal'?
- (5) Line 256: The area (BA25 vs FPI/BA46; BA24 vs FPI/BA46; FPM vs FPI/BA46) by group interactions should be reported (at least in Supplementary).
- (6) Line 285: $p(88) = 0$ -> decimals and exact p value missing.
- (7) Line 332: 'private convictions' -> awkward phrasing, consider rephrasing.
- (8) Line 374: 'extend to BA10 (FPM in humans)' -> Given that both FPM and FPI are in BA10 in humans, consider rephrasing this for clarity.
- (9) Line 387: I think the gender imbalance concern across samples (raised by two reviewers) needs to be framed much more transparently in the Discussion, as 'A limitation of the current study is (...)', as opposed to 'It could be argued that the current findings are biased by the skewed distribution of males and females across the high-anxious and non-anxious groups'.

Reviewer #3:

Remarks to the Author:

The authors have carefully addressed all my concerns, and I have no more comments.

Donders Institute

for Brain, Cognition and Behaviour

**Centre for
Cognitive Neuroimaging**

Bob Bramson
Kapittelweg 29
6500 HB Nijmegen
P.O. Box 9101
6525 EN Nijmegen
The Netherlands

Editorial Board – *Nature Communications*

Dr. Daniel Barry
Associate-editor

+31(0)24 36 22400
bob.bramson@donders.ru.nl
www.ru.nl/donders

Subject: Manuscript Revision Submission

Nijmegen, June 18, 2023

Dear Dr. Barry,

We are delighted to hear that our manuscript has been accepted, in principle, for publication in *Nature Communications*. We are pleased to read that the reviewers are mostly content with the changes we made to the manuscript. In response to remaining concerns, we have added a recognition in the discussion that the inclusions of only male participants in the non-anxious group is a limitation, we have made the requested changes to clarify terminology and we have expanded discussion of the importance of subjective feelings in anxiety.

We hope that, in the light of this revision, our manuscript can be published in *Nature Communications*.

Sincerely,

Bob Bramson, Sjoerd Meijer, Annelies van Nuland, Ivan Toni, Karin Roelofs

Reviewer #1 (Remarks to the Author):

I am satisfied with most of the changes. However, I suggest that the authors include more citations to the subjective feelings of anxiety, such as some of the ones I suggested or others.

We have now added (line 457): “Although we cannot speak to the subjective components of anxiety from our data, it becomes relevant to test how the structure-function relationships we observe relate to individual subjective experiences, given the proposed role of FPI in the consciousness of emotion^{49,50}.”

With 49: LeDoux, J.E. (2020). Thoughtful feelings. *Curr. Biol.* 30, R619–R623.10.1016/j.cub.2020.04.012.

And 50: Taschereau-Dumouchel, V., Michel, M., Lau, H., Hofmann, S.G., and LeDoux, J.E. (2022). Putting the “mental” back in “mental disorders”: a perspective from research on fear and anxiety. *Mol. Psychiatry* 27, 1322–1330. 10.1038/s41380-021-01395-5.)

Reviewer #2 (Remarks to the Author):

I am reviewing the first revision of the paper “Anxious individuals shift emotion control from lateral frontal pole to dorsal prefrontal cortex; functional, structural and neurochemical evidence” by Bramson et al. The authors have addressed most of my concerns, but a few points of concern remain, which are detailed below:

(1) Lines 139-146: Using a test of the null hypothesis in the absence of a significant Group X Congruency interaction (Figure 2C) seems unusual. That paragraph would flow better if the authors first detailed the nature of the FPI effect in the non-anxious group, and then tested the interaction (and the Bayesian null).

We thank the reviewer for this suggestion and have altered this text accordingly.

(2) The lack of interaction (differences in slopes) for results now reported in lines 158-159 & 162-163 is concerning and reduces their interpretability. Apart from “ (...)those 156 participants that recruited FPI the least, relied most on dPFC, $\rho(91)=-.22$, $r = .038$, again suggesting that dPFC compensates for reduced recruitment of FPI”, I would suggest removing them to avoid unwarranted interpretation of group differences in the absence of slope differences. (Reporting that they are n.s. would suffice). Clarify what $b = .11$ [.02 .19] reflects.

We agree; we have removed the sentence indicated and clarified the statistics term by stating:

“Behavioral congruency was also differentially related to neural congruency in dPFC between groups $b =$

.11 [.02 .19]. Namely, in the high-anxious group, neural congruency effects in dlPFC correlated negatively with behavioral congruency; $\rho(50) = -.28$, $p = .04$, whereas this is not the case for the non-anxious participants, $\rho(39) = 0.0$, $p = .98$ ”.

(3) Line 330: Missing word: “(...) the overexcitable FPI we observe in high-anxious (?...)”

We thank the reviewer for pointing this out; the sentence now reads:

Putatively, the combination of overexcitable FPI and stronger amygdala afferences when controlling emotional actions (observed in the high-anxious group), might make it difficult for anxious individuals to maintain their private sense of confidence in their opinions when conforming to social norms, a role attributed to FPI⁴².

(4) Nomenclature re: dorsal prefrontal cortex-> given this is referring to 9-46, wouldn't it be more anatomically specific to refer to this region as dorsolateral PFC (DLPFC) instead of just 'dorsal'?

We have changed the nomenclature from dorsal to dorsolateral prefrontal cortex (dlPFC) throughout the manuscript.

(5) Line 256: The area (BA25 vs FPI/BA46; BA24 vs FPI/BA46; FPM vs FPI/BA46) by group interactions should be reported (at least in Supplementary).

We have now added the tests between groups for area 24, 25 and FPM to the supplementary materials (page 5).

(6) Line 285: $p(88) = 0$ -> decimals and exact p value missing.

Exact decimals and p-value have been added: $p(88) = 0.0$, $p = .99$.

(7) Line 332: 'private convictions' -> awkward phrasing, consider rephrasing.

We thank the reviewer for pointing this out, the sentence now reads:

(8) Line 374: 'extend to BA10 (FPM in humans)' -> Given that both FPM and FPI are in BA10 in humans, consider rephrasing this for clarity.

This now reads:

It is possible that the projections to FPI are an extension of magnocellular basal nucleus projections that, in macaque, extend to the medial part of BA10 (FPM in humans)⁶⁰

(9) Line 387: I think the gender imbalance concern across samples (raised by two reviewers) needs to be framed much more transparently in the Discussion, as 'A limitation of the current study is (...)', as opposed to 'It could be argued that the current findings are biased by the skewed distribution of males and females across the high-anxious and non-anxious groups'.

This now reads:

A limitation of the current study is the inclusion of only male participants in the non-anxious group. However, we consider it unlikely that the neural and behavior congruency effects can be solely attributed to this factor. Namely, both male-only^{3,30} and female only⁶³ studies using the AA task have shown FPI recruitment, and large scale mixed samples did not find differences between males and female participants in FPI engagement^{9,64,65}.